# CONCRETIZER: MODEL INVERSION ATTACK VIA OCCUPANCY CLASSIFICATION AND DISPERSION CONTROL FOR 3D POINT CLOUD RESTORATION

∗**Youngseok Kim**[1]     ∗**Sunwook Hwang**[2§]     †**Hyung-Sin Kim**[3]     †**Saewoong Bahk**[1]
[1]Department of Electrical and Computer Engineering, Seoul National University
[2]System LSI, Samsung Electronics
[3]Graduate School of Data Science, Seoul National University
`yskim@netlab.snu.ac.kr, sunw.hwang@samsung.com,`
`{hyungkim, sbahk}@snu.ac.kr`

## ABSTRACT

The growing use of 3D point cloud data in autonomous vehicles (AVs) has raised serious privacy concerns, particularly due to the sensitive information that can be extracted from 3D data. While model inversion attacks have been widely studied in the context of 2D data, their application to 3D point clouds remains largely unexplored. To fill this gap, we present the first in-depth study of model inversion attacks aimed at restoring 3D point cloud scenes. Our analysis reveals the unique challenges, the inherent sparsity of 3D point clouds and the ambiguity between empty and non-empty voxels after voxelization, which are further exacerbated by the dispersion of non-empty voxels across feature extractor layers. To address these challenges, we introduce *ConcreTizer*, a simple yet effective model inversion attack designed specifically for voxel-based 3D point cloud data. *ConcreTizer* incorporates Voxel Occupancy Classification to distinguish between empty and non-empty voxels and Dispersion-Controlled Supervision to mitigate non-empty voxel dispersion. Extensive experiments on widely used 3D feature extractors and benchmark datasets, such as KITTI and Waymo, demonstrate that *ConcreTizer* concretely restores the original 3D point cloud scene from disrupted 3D feature data. Our findings highlight both the vulnerability of 3D data to inversion attacks and the urgent need for robust defense strategies.

## 1 INTRODUCTION

Recent advancements in Autonomous Vehicles (AVs) have underscored the importance of continuous vision data collection and sharing. At the same time, the widespread adoption of AI technology has amplified privacy concerns, prompting increased research on this issue (Guo et al., 2017; Stahl & Wright, 2018). Consequently, AV's data collection faces strict regulations that requires data de-identification (Mulder & Vellinga, 2021). For example, the EU's General Data Protection Regulation (GDPR) (EU, 2016) mandates businesses to adopt stringent data protection protocols.

Beyond these regulations, the need for privacy preservation is rapidly increasing, particularly in 3D point cloud data. This is because various types of privacy-related information can be revealed through rich 3D shape information. For instance, personal identities can be exposed through facial recognition (Zhang et al., 2019) and person re-identification (Cheng & Liu, 2021). Additionally, behavioral patterns can be inferred from human pose estimation (Zhou et al., 2020) and activity recognition (Singh et al., 2019b). Location information can also be extracted using techniques like Simultaneous Localization and Mapping (SLAM) (Kim et al., 2018). Furthermore, the ability to reconstruct 2D images from sparse 3D data (Pittaluga et al., 2019; Song et al., 2020) emphasizes the importance of securing raw 3D point data from the outset.

---

∗ Both authors contributed equally to this research.
§ This work was conducted while the author was affiliated with Seoul National University.
† Corresponding authors.

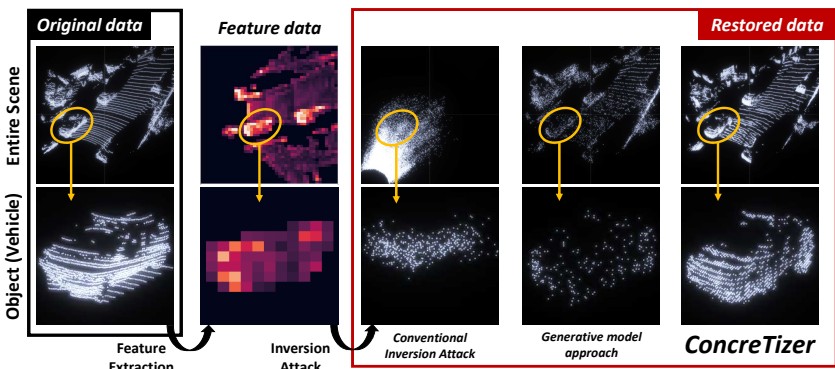

Figure 1: **Inversion attack results of a 3D point cloud.** Feature data is extracted from original point cloud through a 3D feature extractor (Yan et al., 2018). *ConcreTizer* (right) enables restoration with simple modifications to conventional approach (left), and even achieves more concrete restoration than generative model approach (middle) (Xiong et al., 2023).

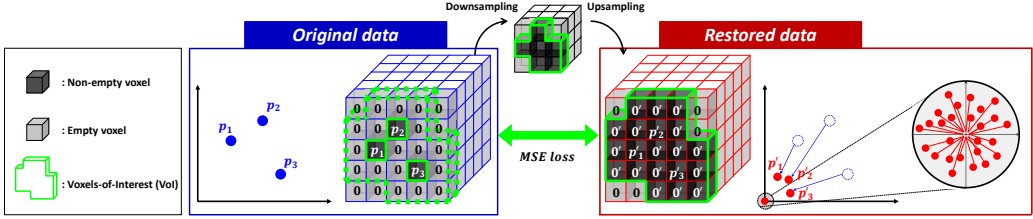

Figure 2: **Restoration through conventional inversion attack method.** Voxelization introduces zero-padding to empty voxels. During downsampling and upsampling, non-empty voxels spread to neighboring areas, expanding the VoI (green region). Within the VoI, voxel-wise channel regression generates additional points in zero-padded regions, leading to clustering near the origin.

However, research on privacy in 3D point cloud data remains significantly underexplored compared to advancements in the 2D image domain. A prominent research area in 2D image privacy is **inversion attack**, which aims to restore the original data from extracted feature. While earlier studies (Gupta & Raskar, 2018; Vepakomma et al., 2018; Singh et al., 2019a) suggested that 2D images could be anonymized by extracting features, inversion attacks have demonstrated that these features can be used to restore the original 2D images. In contrast, while there have been a few prior studies on privacy of 3D data (Wang et al., 2024a), inversion attacks on 3D data remain largely unexplored. This research gap allowed a recent study (Hwang et al., 2023) to operate under the assumption that disseminating 3D features inherently prevents the restoration of the original data. In the absence of existing inversion attack methods for 3D data, the authors developed a Point Regression method to invert voxel-based backbones, aiming to demonstrate that restoring the original 3D scene from its extracted features is infeasible. As in Figure 1, the conventional Point Regression in (Hwang et al., 2023) fails to restore 3D point cloud data from intermediate features.

We argue that this failure is not due to an inherent safety of 3D features but rather a lack of careful design that considers the characteristics of 3D backbones. To address this issue, Figure 2 examines the phenomena arising when the Point Regression method inverts voxel-based feature extractors, which are dominant architectures in autonomous driving applications. The Point Regression approach attempts to directly restore point coordinates within each voxel by minimizing mean squared error (MSE). The problem is as follows: The sparsity of 3D point cloud data results in a large number of zero-padded voxels. To identify the meaningful regions within the voxel grid, we define VoI (Voxels-of-Interest) as the set of non-empty voxels, which contain valuable information. During both feature extraction and inversion processes, VoI spread into empty voxels. This dispersion leads to a proliferation of false VoI (originally empty voxels), causing Point Regression to erroneously generate points in regions that were initially void. Moreover, these false VoI disproportionately impact the MSE loss, prompting the Point Regression model to bias the restoration by concentrating most points near the origin (0, 0, 0) to minimize estimation errors for the false VoI. This bias significantly degrades localization performance for the relatively smaller number of true VoI (originally non-empty voxels).

The analysis reveals that the key to a successful inversion attack is not restoring the representation of the voxel (i.e., point coordinates) but accurately determining whether a voxel was originally empty or non-empty. Once this classification is achieved, localizing points within non-empty voxels becomes more straightforward, as the error is constrained by the typically small voxel size. Based on this insight, we transform the conventional Point Regression problem into a more explicit Voxel Occupancy Classification (VOC) problem. In addition, the spread of VoI should be suppressed during restoration to minimize the negative impact of false VoI. To address this, our model incorporates Dispersion-Controlled Supervision (DCS), which segments the feature extractor based on downsampling layers and trains each segment individually, proactively controlling the dispersion of VoI. Thanks to its tailored design, our model, *ConcreTizer*, even outperforms the generative model approach that uses conditional generation (see Figure 1, the generative model approach (Xiong et al., 2023)).

To demonstrate the general applicability of *ConcreTizer*, we deployed it on two representative 3D feature extractors (Yan et al., 2018; Lu et al., 2022), which are essential components in various applications including 3D object detection, 3D semantic segmentation, and tracking. Our experiments on the widely used KITTI (Geiger et al., 2012) and Waymo (Sun et al., 2020) datasets confirm that *ConcreTizer* consistently outperforms across various datasets and 3D feature extractors. We showcase the superior performance of *ConcreTizer* through a comprehensive set of quantitative and qualitative evaluations, including point cloud similarity metrics, visual analysis, task-specific performance (3D object detection) using restored scenes, and the effectiveness of potential defense mechanisms.

The contributions of this paper are as follows:

- This is the first in-depth study on model inversion attacks for restoring voxel-based 3D point cloud scenes, identifying unique challenges from the interaction between sparse point clouds and voxel-based feature extractors.
- To address the identified challenges, we propose *ConcreTizer*, tailored for inverting 3D backbone networks, with Voxel Occupancy Classification and Dispersion-Controlled Supervision.
- Through extensive experiments with representative 3D feature extractors and well-established open-source datasets, we demonstrate the effectiveness of *ConcreTizer* in both quantitative and qualitative aspects.

## 2 RELATED WORK

**3D Point Clouds Feature Extraction.** Feature extractors for 3D point cloud data encompass set, graph, and grid-based approaches, each distinguished by its representation format. The computational complexity of set and graph-based methods (Qi et al., 2017; Kipf & Welling, 2016; Park et al., 2023) scales significantly with the number of points, limiting their use in real-time applications like autonomous driving. Conversely, grid-based methods (Zhou & Tuzel, 2018; Yan et al., 2018; Shi et al., 2020; Sun et al., 2022) organize the 3D space into a voxel grid and apply specialized convolution (Liu et al., 2015; Graham & Van der Maaten, 2017) for efficient feature extraction from sparse data. This efficiency makes them particularly well-suited for autonomous driving applications. Based on these characteristics, we investigate inversion attacks for scenarios using voxel-based feature extractors.

**Model Inversion.** Model inversion was originally explored in the context of interpreting deep learning models. Traditional approaches generate saliency maps to understand how models produce outputs (Du et al., 2018). Other methods (Mahendran & Vedaldi, 2015; Dosovitskiy & Brox, 2016b;a) reconstruct the input from intermediate features to analyze the information flow through model layers. Recently, with growing concerns about data privacy, model inversion has gained attention as a privacy attack. Early studies attempted to restore input face images from confidence scores (Yang et al., 2019b). Subsequent studies (Zhang et al., 2020; Zhao et al., 2021) leverage additional information for more sophisticated restoration. Building on these studies, corresponding defense techniques (Liu et al., 2019; Xue et al., 2023; Dusmanu et al., 2021; Ng et al., 2022; Zhang et al., 2022) have also been investigated, enriching the exploration of data privacy. However, existing work has primarily focused on 2D image data. There is a clear need for an inversion attack technique that accounts for the unique characteristics of 3D point cloud data in autonomous driving. To the best of our knowledge, this research is the first to study inversion attacks on 3D data.

**Point Cloud Generation.** Generative models are widely used, owing to their diverse range of applications. In the 3D point cloud domain, several generative models are actively being explored. Unconditional generation tasks aim to create plausible 3D shapes from random inputs, such as

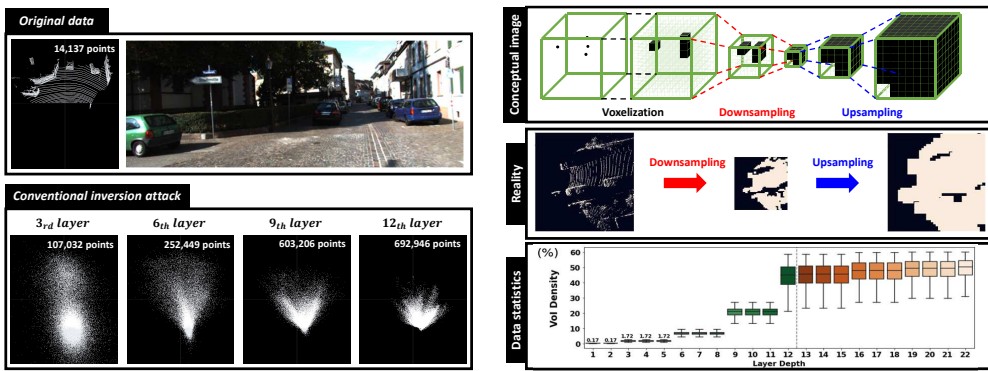

Figure 3: **(Left) The results of the conventional inversion attack:** As the layer depth increases, the number of restored points increases rapidly, and the concentration of points near the origin becomes more noticeable. **(Right) The VoI (Voxels-of-Interest) dispersion effect:** The non-empty voxels spread as they pass through the feature extractor and inversion attack model.

noise (Achlioptas et al., 2018; Valsesia et al., 2018; Yang et al., 2019a; Luo & Hu, 2021). Conditional generation tasks involve generating the missing part of a point cloud (Yu et al., 2021; Huang et al., 2020; Wen et al., 2020) or producing a 3D point cloud from a 2D image (Mandikal et al., 2018; Mandikal & Radhakrishnan, 2019; Melas-Kyriazi et al., 2023). However, most existing research focuses on dense point cloud data for individual objects (e.g., Chang et al. (2015)). Only a few studies (Caccia et al., 2019; Zyrianov et al., 2022) deal with scene-level sparse point clouds captured from autonomous vehicles. Even these studies require specific representation formats and do not support using 3D grid-type features, as conditions in our inversion attack scenario. To our knowledge, the only scene-level sparse point cloud generation model based on 3D grid representations is Xiong et al. (2023). We also conducted performance comparisons with conditional generation approach.

## 3 PRELIMINARY: LIMITATIONS OF CONVENTIONAL INVERSION ATTACK

The only known attempt at an inversion attack on 3D point cloud data is by Hwang et al. (2023). Even this research does not directly focus on inversion attacks but rather seeks to assess the privacy protection effectiveness of 3D features by developing a simple inversion attack based on Point Regression. Before designing our method, we explore why the conventional approach can not effectively restore 3d point cloud scenes (Figure 3, left).

Firstly, we identified an issue in voxel-based models related to the voxelization process. During voxelization, regions without points are zero-padded. However, conventional regression method does not consider point existence but focus solely on point localization, mistakenly interpreting zero-padded representations as valid points located at (0, 0, 0). As a result, points are created even for empty voxels, leading to an overgeneration of points compared to the original data. Secondly, the inherently sparse nature of point clouds results in a large number of zero-padded voxels, far exceeding those containing valid points. Since Point Regression-based inversion attacks aim to minimize estimation errors across all voxels, they unintentionally prioritize zero-padded regions. Consequently, this bias towards zero-padded voxels causes an over-concentration of points near the origin in the restored scene. Moreover, it significantly increases localization errors for the relatively smaller number of valid points, as these errors become negligible within the overall regression error.

Lastly, we observed that as the feature extractor layers deepen, existing attack methods are increasingly hindered by the negative impact of zero-padded voxels: (1) an excessive number of restored points and (2) an intensified concentration of points near the origin. Specifically, if voxels with a value of (0, 0, 0) persist in the final restored state, they are excluded from the regression targets and do not directly affect the regression loss. However, due to the nature of convolution operations, the values of non-empty voxels—defined as VoI (Voxels-of-Interest)—gradually disperse into the surrounding empty voxels. As the layers deepen, more originally zero-padded voxels become non-empty during the feature extraction and inversion processes. Consequently, an increasing number of these previously zero-padded voxels are included in the regression targets. Our experiments revealed that the density of VoI spikes significantly at downsampling layers (Figure 3, right), further amplifying the influence of zero-padded voxels on the final restoration results.

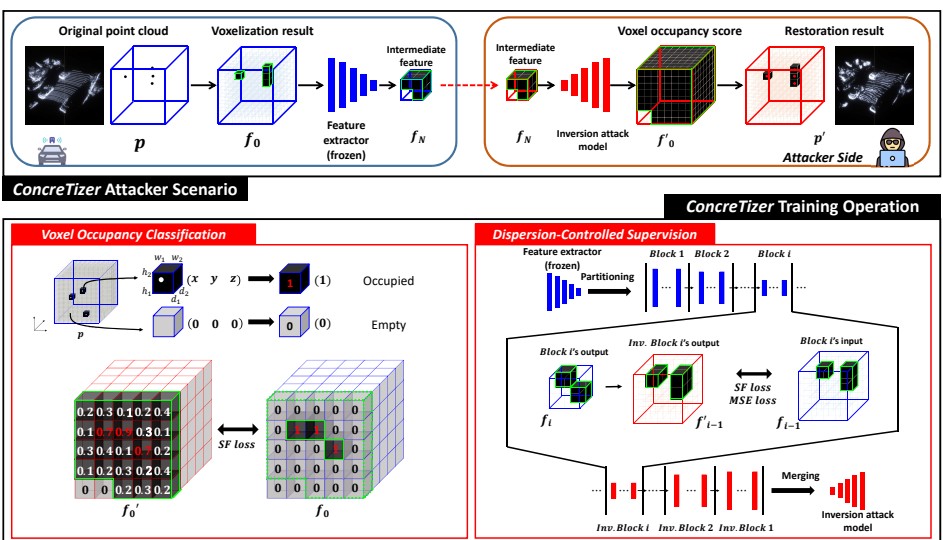

Figure 4: *ConcreTizer* **framework.** Original point cloud and features are denoted as $p$ and $f_i$, with restored versions as $p'$ and $f'_i$, respectively, where $i$ indicates the $i$-th downsampling layer. *ConcreTizer* restores data by classifying $f_0$'s occupancy and placing points at voxel centers. For deeper layers, it partitions at downsampling layers to restore $f_{i-1}$ from $f_i$.

## 4 PROPOSED METHOD

### 4.1 AV SCENARIO

We focus on autonomous vehicle (AV) scenarios due to their high risk of exposure to inversion attacks. In AV contexts, feature data would be shared for purposes such as computation offloading (Xiao et al., 2022; Hanyao et al., 2021), model enhancement (Hwang et al., 2023), and cooperative inference (Wang et al., 2020; Xu et al., 2022; Yu et al., 2022). Specifically, we selected voxel-based feature extractors, which are well-suited for real-time processing in AV. Their efficiency makes them essential for tasks such as 3D object detection (Yan et al., 2018; Lang et al., 2019; Shi et al., 2019; 2020; Shi & Rajkumar, 2020), semantic segmentation (Wu et al., 2019; Thomas et al., 2019), and tracking (Yin et al., 2021). In this scenario, an attacker with access to the same feature extractor can easily prepare 3D point cloud data for training the inversion attack model. Since the restoration task doesn't require separate labeling, they can utilize open-source datasets or self-collected data.

### 4.2 PROBLEM DEFINITION

The goal of an inversion attack is to discover the inverse process of a given feature extractor in order to restore the original data. For voxel-based feature extractors, the initial step involves a voxelization process that transforms point cloud data into a grid format. Voxelization converts a 3D point cloud $p \in \mathbb{R}^{k \times 3}$, where $k$ is the number of points, into a voxel grid $f_0 \in \mathbb{R}^{3 \times H \times W \times D}$, where $H, W, D$ represent the spatial dimensions of the grid. The x, y, and z coordinate information is organized into separate channels, and voxels without points are zero-padded, resulting in channel values of (0, 0, 0). In particular, during the downsampling process, the spatial dimensions shrink while the channel size increases, producing features $f_N \in \mathbb{R}^{C_N \times h_N \times w_N \times d_N}$, where $N$ is the number of downsampling layers, $C_N > 3$, and $h_N, w_N, d_N$ are smaller than $H, W, D$. Consequently, our inversion attack aims to restore the original voxel grid $f_0$ from the downsampled features $f_N$.

### 4.3 CONCRETIZER FRAMEWORK

Figure 4 depicts the overall *ConcreTizer* framework incorporating the scenario and attacker-side training operations. For the design of the inversion attack model, we adopted a symmetrical structure to the feature extractor, following previous studies (Yang et al., 2019b; Zhang et al., 2020; Zhao et al., 2021). In this approach, the original shape is restored by upsampling at the positions where downsampling occurred (detailed structure is provided in the supplementary material). Building upon symmetric structure, *ConcreTizer* applies Voxel Occupancy Classification (VOC) and Dispersion-Controlled Supervision (DCS) to overcome the limitations of traditional inversion attack. VOC converts the regression problem into a classification problem to address the issue of point clustering near the origin. DCS prevents the dispersion of VoI by splitting the feature extractor, helping to mitigate the degradation of restoration performance as the network deepens.

### 4.3.1 VOXEL OCCUPANCY CLASSIFICATION

In traditional inversion attack methods, the original data is directly restored through regression on channel values. In our scenario, since the x, y, and z coordinates are channelized during the voxelization process, performing regression would restore coordinate values. However, since voxelization of sparse point clouds produces a large number of zero-padded voxels with (0, 0, 0) channel value, many unnecessary points cluster near the origin in the inversion attack results (Figure 3, left). To address this issue, we transform the regression problem into a classification problem to resolve the semantic ambiguity of zero-padded voxels—whether they represent empty voxels or valid points at coordinates (0, 0, 0). This can be achieved through simple binary encoding, where each voxel is labeled as 0 (*negative occupancy*) or 1 (*positive occupancy*), making the meaning of zero-padding clear. Using the VOC method, the inversion attack model outputs binary classification scores in the form of $\mathbb{R}^{1 \times H \times W \times D}$, rather than continuous coordinate values in the form of $\mathbb{R}^{3 \times H \times W \times D}$. If a voxel is determined to contain a point, the corresponding coordinate can be restored easily. This is because the range of coordinate values is bounded by the spatial location of the voxel, and the voxel size is typically small enough. As a result, by using the center coordinates of the voxel, we achieve effective restoration within an error range constrained by the voxel size.

Additionally, due to the sparsity of original point cloud data, the binary-encoded labels contain a higher ratio of 0s compared to 1s. This phenomenon is particularly exacerbated as the depth of the layers increases. Let $f_0$ be the original voxelized point cloud and $f_0'$ be the restored one by the inversion attack. The number of positive labels is fixed as |VoI of $f_0$|, while the number of negative labels, |VoI of $f_0'$| − |VoI of $f_0$|, increases exponentially as the depth of the layer increases. To account for this imbalance, we apply the Sigmoid Focal (SF) loss (Lin et al., 2017), a variant of the conventional cross-entropy loss. The mathematical representation of the SF loss is given by $\text{FL}(p_t) = -\alpha_t(1 - p_t)^\gamma \log(p_t)$, where $p_t$ denotes the model's predicted probability for the target class. The factor $\alpha_t$ is employed to adjust the importance given to the positive and negative classes.

### 4.3.2 DISPERSION-CONTROLLED SUPERVISION

While applying SF loss in VOC can partially address the label imbalance issue, it cannot prevent the more inherent problem of VoI dispersion. The original data is sparse with many empty voxels, yet as observed earlier, the VoI density increases exponentially during the downsampling process (Figure 3, right). As the VoI spreads excessively in the deeper layers, it becomes increasingly difficult to restore the data to its original sparse state.

Our proposed DCS offers a more fundamental solution to address VoI dispersion. It divides the feature extractor into multiple *blocks* and performs restoration progressively. First, the feature extractor is partitioned based on the downsampling layer, where VoI dispersion occurs. In the inversion attack model, a corresponding *inversion block* is created for each *block* of feature extractor. This allows the restoration process to be trained in block units, effectively controlling VoI dispersion within each block. It is important to note that, at the original voxel level, the channel values directly represent point coordinates, eliminating the need for regression (if the classification result is positive, the channel value is estimated as the center coordinate of the voxel). However, at the intermediate feature level, normalization is applied, which disrupts the direct relationship between the channel values and the voxel location. As a result, both classification and regression on the channel values are required.

For example, if the input to the $(i + 1)$-th *block* is $f_i \in \mathbb{R}^{C_i \times h_i \times w_i \times d_i}$ and the output is $f_{i+1} \in \mathbb{R}^{C_{i+1} \times h_{i+1} \times w_{i+1} \times d_{i+1}}$, then the $(i + 1)$-th *inversion block* in the inversion attack model takes $f_{i+1}$ as input and produces $f_i' \in \mathbb{R}^{C_i \times h_i \times w_i \times d_i}$, which is the result of restoring $f_i$. Specifically, $m_i' \in \mathbb{R}^{1 \times h_i \times w_i \times d_i}$ (spatial occupancy scores found by applying SF loss) and $c_i' \in \mathbb{R}^{C_i \times h_i \times w_i \times d_i}$ (channel values found by applying L2 loss) are derived from $f_{i+1}$. Then, $c_i'$ is masked by using $m_i'$ to generate $f_i'$. During the masking process, unnecessary voxel values are erased, helping to suppress the dispersion of VoI. Note that in the first *inversion block*, which is the final stage of the inversion attack model, only classification is performed, with no additional regression. The loss function for each *inversion block* is:

$$Loss(inversion\ block\ i + 1) = \begin{cases} L_{\text{cls}} & \text{if } i = 0, \\ L_{\text{cls}} + \beta \cdot L_{\text{reg}} & \text{if } i \geq 1. \end{cases}$$

$$L_{\text{cls}} = \sum_{\text{VoI}} \text{SF loss}(m_i, m_i') \quad \text{and} \quad L_{\text{reg}} = \sum_{\text{VoI}} \text{L2 loss}(c_i, c_i')$$

Table 1: **Inversion attack result with KITTI and Waymo dataset.** Average CD and HD values in centimeters, and F1 scores with 15 cm and 30 cm thresholds for KITTI and Waymo datasets. Metrics evaluate over each dataset with 3769 and 3999 scenes, respectively.

| #Downsampling (LayerDepth) | | 1 (3rd) | | | 2 (6th) | | | 3 (9th) | | | 4 (12th) | | |
|---|---|---|---|---|---|---|---|---|---|---|---|---|---|
| | | CD (↓) | HD (↓) | F1score (↑) | CD (↓) | HD (↓) | F1score (↑) | CD (↓) | HD (↓) | F1score (↑) | CD (↓) | HD (↓) | F1score (↑) |
| KITTI | Point Regression | 1.3868 | 23.5855 | 0.3543 | 1.2879 | 34.2395 | 0.3904 | 3.1229 | 54.0173 | 0.2110 | 4.1439 | 56.9811 | 0.1298 |
| | UltraLiDAR | 0.0744 | 8.2269 | 0.9122 | 0.0818 | 8.0974 | 0.8905 | 0.0836 | 7.9561 | 0.8869 | 0.1012 | **7.9185** | 0.8152 |
| | *ConcreTizer* | **0.0321** | **7.5603** | **0.9918** | **0.0373** | **7.5249** | **0.9914** | **0.0507** | **7.8453** | **0.9793** | **0.0776** | 8.1193 | **0.9160** |
| Waymo | Point Regression | 1.4979 | 55.6589 | 0.7644 | 2.7733 | 66.7899 | 0.6489 | 4.1053 | 70.6608 | 0.5524 | 4.9340 | 71.9608 | 0.4355 |
| | UltraLiDAR | 0.0810 | 10.9582 | 0.9742 | 0.0898 | 11.3360 | 0.9623 | 0.1017 | 11.4987 | 0.9503 | 0.1378 | 12.0259 | 0.8849 |
| | *ConcreTizer* | **0.0374** | **10.2544** | **0.9984** | **0.0466** | **10.2326** | **0.9979** | **0.0712** | **10.5724** | **0.9781** | **0.1087** | **11.3399** | **0.9251** |

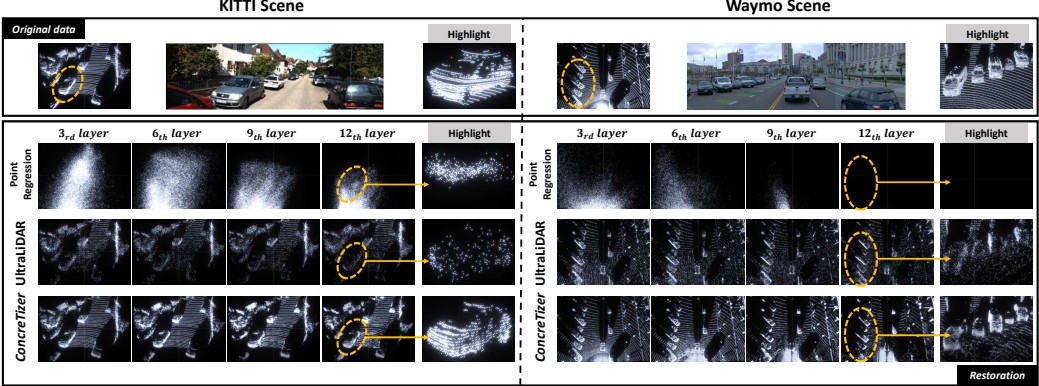

Figure 5: **Qualitative results for KITTI (scene 73) and Waymo (scene 79).** Top shows the original point cloud, 2D image, and highlighted region. Below, restoration performance of three techniques is displayed, progressing left to right by layer depth.

Here, $m_i$ and $m_i'$ represent the ground truth and predicted spatial occupancy masks, respectively, while $c_i$ and $c_i'$ denote the ground truth and predicted channel values. The final result of passing through all *inversion blocks* is a set of binary classification scores in the form of $\mathbb{R}^{1 \times H \times W \times D}$. Restoration is completed by generating a point at the center of the voxel corresponding to positive occupancy.

## 5 EXPERIMENTS

### 5.1 EXPERIMENTAL SETUP

**3D Feature Extractor.** We employ voxelization-based 3D feature extractors as the target of our inversion attack. Based on the OpenPCDet (Team, 2020) project, we utilize pre-trained VoxelBackbone (Yan et al., 2018) and VoxelResBackbone (Lu et al., 2022), extensively used in key applications for 3D point cloud data. The VoxelBackBone structure includes four downsampling layers (i.e., $N = 4$), each preceded by two convolutional layers, while the VoxelResBackbone incorporates additional convolutional layers and skip connections.

**Inversion Model Training.** We train the inversion attack model on the real-world KITTI (Geiger et al., 2012) and Waymo (Sun et al., 2020) datasets. In VOC, when applying the SF loss function, only $\alpha$ in the hyperparameters is adjusted. In DCS, the weight on the regression loss, $\beta$, is set to 1.

**Metrics.** To evaluate 3D scene restoration performance, we employ various metrics. For qualitative analysis, we visualize the 3D point cloud using the KITTI viewer web tool. For quantitative analysis, we utilize point cloud similarity metrics such as Chamfer Distance (CD) (Borgefors, 1984), Hausdorff Distance (HD) (Huttenlocher et al., 1993), and F1 Score (Goutte & Gaussier, 2005). Additionally, to assess the utility of the restored data, we examine 3D object detection accuracy using pre-trained detection models.

### 5.2 RESTORATION PERFORMANCE

**Comparison Schemes.** To demonstrate the superiority of *ConcreTizer*, we compare it with two approaches: a traditional inversion attack method and a generative model-based approach. First, we examine Point Regression (Mahendran & Vedaldi, 2015; Dosovitskiy & Brox, 2016a;b), a conventional inversion attack method. In this approach, the goal is to directly recover the channel

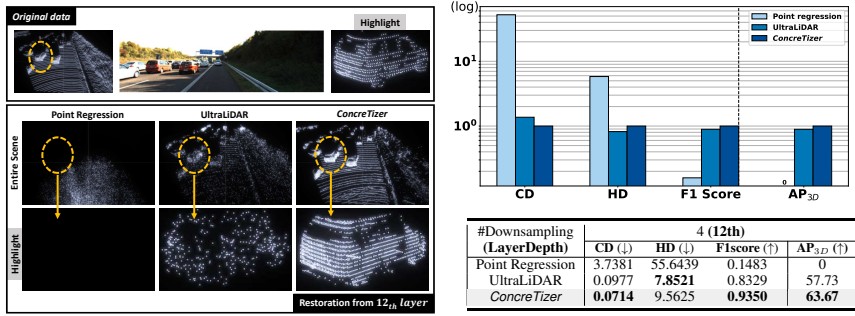

Figure 6: **Restoration result on VoxelResBackbone with KITTI dataset.** At the left, the last layer's restoration performance for three techniques is shown. At the right, average performance across the KITTI dataset is presented. A bar graph depicts relative performance, and a table details raw values.

values. To improve the results, we additionally apply post-processing to remove points that fall outside the defined point cloud range or cluster excessively near the origin. Next, we compare *ConcreTizer* with a generative model-based approach. Inversion attacks using generative models require conditional generation, where feature data serve as the condition. Among existing LiDAR point cloud generation models, UltraLiDAR (Xiong et al., 2023) is the only one utilizing a voxel representation similar to our feature extractor. To adapt UltraLiDAR for inversion, we modified its encoder to accept voxel features as input.

**Result Analysis.** Table 1 presents the point scene restoration performance at different layer depths of VoxelBackBone (Yan et al., 2018), while Figure 5 visualizes the corresponding restored point cloud scenes. It is evident that *ConcreTizer* consistently demonstrates outstanding performance across all cases in both the KITTI and Waymo datasets.

Traditional Point Regression methods prove ineffective for inversion attacks on 3D features. In particular, many points cluster near the origin, and this phenomenon becomes more pronounced at deeper layers. This limitation stems from the failure to account for the characteristics of 3D sparse features. By leveraging conditional generation, UltraLiDAR can restore the overall scene in a coarse-grained manner, showing less performance degradation in terms of the HD metric as layer depth increases. This rough recovery can be attributed to the transformation of 3D sparse features into 2D dense features, which aligns with the 2D VQ-VAE design. Since VoI dispersion is no longer present in 2D dense features, UltraLiDAR achieves better stability. However, this conversion leads to the loss of 3D sparse characteristics, resulting in less accurate restoration of fine details. In contrast, *ConcreTizer* effectively suppresses VoI dispersion through DCS while preserving the sparse nature of 3D features. Despite its simple design, it achieves more concrete restoration compared to the generative model-based approach. At the deepest layer, *ConcreTizer* outperforms the generative approach by 23.4% and 12.4% on KITTI, and by 21.1% and 4.5% on Waymo in terms of CD and F1 score, respectively.

Additionally, Figure 6 presents results for VoxelResBackbone (Lu et al., 2022). When analyzing the representative results from the deepest layer, *ConcreTizer* exhibits the best performance in CD, F1 Score, and $AP_{3D}$. A persistent limitation of UltraLiDAR is the lack of detailed shape in the inversion attack result. Detailed experimental results, including those for VoxelResBackbone and the Waymo dataset, are provided in the supplementary materials.

## 5.3 ATTACK PERFORMANCE IN THE CONTEXT OF 3D OBJECT DETECTION

To assess the effectiveness of inversion attack results in terms of privacy compromise, we measure the 3D object detection accuracy using restored point cloud scenes with pre-trained object detection models. Table 2 summarizes the benchmark results for the KITTI and Waymo datasets. Point Regression fails to perform inversion attack, producing completely unusable results. UltraLiDAR performs relatively well on KITTI but exhibited poor performance on Waymo, which has a broader range and higher scene complexity. This suggests that while generative models can restore overall scene, they struggle to capture detailed shape. In contrast, only *ConcreTizer* demonstrates consistent performance across both datasets, achieving 75.5 to 87.0% and 62.6 to 75.7% of the detection performance compared to the original scenes in KITTI and Waymo, respectively.

Table 2: **3D object detection results with KITTI and Waymo datasets.** The reported metric for the KITTI dataset is Average Precision (AP) at hard difficulty, while for the Waymo dataset, Average Precision weighted by Heading (APH) is reported at LEVEL2 difficulty.

| | Detection Model | PointPillar | PVRCNN | VoxelRCNN | PointRCNN |
|---|---|---|---|---|---|
| KITTI | Original Data | 76.11 | 78.82 | 78.78 | 78.25 |
| | Point Regression | 0 | 0 | 0 | 0 |
| | UltraLiDAR | 58.32 | 56.08 | 59.00 | 54.19 |
| | *ConcreTizer* | **66.25** | **59.48** | **64.27** | **65.03** |

| | Detection Model | PointPillar | PVRCNN | VoxelRCNN | CenterPoint |
|---|---|---|---|---|---|
| Waymo | Original Data | 0.5604 | 0.6534 | 0.6554 | 0.6239 |
| | Point Regression | 0 | 0 | 0 | 0 |
| | UltraLiDAR | 0.2328 | 0.1602 | 0.2179 | 0.1944 |
| | *ConcreTizer* | **0.4245** | **0.4369** | **0.4100** | **0.4107** |

Figure 7: **Ablation study on VoxelBackbone with KITTI dataset.** At the left, the restoration performance for three cases is shown. At the right, average performance across the KITTI dataset is presented wih boxplot.

## 5.4 ABLATION STUDY: COMPONENT-WISE ANALYSIS

To understand the performance of *ConcreTizer*, we analyze the impact of each component. Figure 7 compares the performance of VOC (BCE loss), VOC, and *ConcreTizer* (VOC+DCS). Firstly, VOC (BCE loss) shows that transitioning from regression to classification, which clarifies the meaning of zero-padded voxels, enables restoration of 3D sparse data ($6_{th}$ layer result). However, BCE loss struggles with significant label imbalance as layer depth increases. Comparing VOC (BCE loss) with VOC highlights that SF loss helps alleviate the label imbalance issuet. Nonetheless, in VOC, the restored points cluster in specific areas, leading to biased restoration ($12_{th}$ layer result). Only *ConcreTizer* successfully restores points in a distribution closely matching the original data. This success stems from DCS's ability to effectively mitigate VoI dispersion, especially in deeper layers.

## 5.5 PARTITIONING POLICY IN DISPERSION-CONTROLLED SUPERVISION

We conduct experiments to identify the effective strategy for applying DCS in *ConcreTizer*, given a specific 3D feature extractor. Restoration performance is evaluated on the KITTI dataset by varying the number of DCS instances (i.e., the number of *inversion blocks*). In each case, partitioning is applied at positions that aim to achieve an even division of the total number of layers. As shown in Figure 8, applying 10 DCS instances results in significantly worse performance than not using DCS at all (i.e., DCS 1). This is because the restoration error accumulates as it passes through multiple *inversion blocks*. The best performance is achieved with 2, 3, or 4 DCS instances, where each partitioned block contains at least one downsampling layer. This can be attributed to the additional supervision effectively suppressing VoI dispersion that occurs during the downsampling process. Therefore, to maximize the benefits of supervision, partitioning should be aligned with the downsampling layers, where VoI dispersion manifests. Qualitative results for different DCS instances and further discussion on the optimal DCS split position are provided in the supplementary materials.

## 5.6 TRADEOFF BETWEEN PRIVACY AND UTILITY

To analyze the trade-off between utility (3D object detection accuracy) and privacy protection (restoration error), we examine various data perturbation techniques (Wang et al., 2024b; Li et al., 2021; Wang et al., 2024a) as potential defense mechanisms against the *ConcreTizer* inversion attack. We explored two types of perturbations: **point cloud augmentations** and **Gaussian noise addition**. For point cloud augmentations, we apply *random rotations*, *random scaling*, and *random sampling*. For Gaussian noise addition, we introduce noise at the feature data level with three region-specific configurations: *distributed noise*, which is uniformly applied across all feature data regions; *feature-centric noise*, which is applied only to VoI (regions containing information); and *empty-centric noise*, which exclusively targeted empty regions.

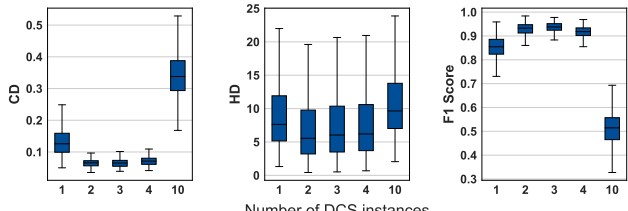

Figure 8: **Effect of the number of DCS instances.** DCS 1 is the end-to-end approach without partitioning. DCS 2, 3, and 4 use downsampling-based partitioning. DCS 10 partitions at every layer.

Table 3: **Effect of point cloud augmentation.** Measured: $AP_{3D}$ (detection accuracy) of the SECOND model and CD (restoration error) of *ConcreTizer*.

| Rotation (°) | 0 | 1 | 2 | 3 | 4 | 5 |
|---|---|---|---|---|---|---|
| AP | 81.77 | 38.38 | 17.08 | 12.10 | 6.10 | 2.78 |
| CD | 0.0776 | 0.1142 | 0.1728 | 0.2310 | 0.2848 | 0.3344 |

| Scaling (%) | 0 | 2 | 4 | 6 | 8 | 10 |
|---|---|---|---|---|---|---|
| AP | 81.77 | 54.12 | 24.09 | 11.47 | 8.88 | 5.26 |
| CD | 0.0776 | 0.1468 | 0.2253 | 0.2780 | 0.3179 | 0.3516 |

| Sampling (%) | 100 | 25 | 20 | 15 | 10 | 5 |
|---|---|---|---|---|---|---|
| AP | 81.77 | 63.35 | 58.32 | 52.71 | 40.31 | 24.59 |
| CD | 0.0776 | 0.1368 | 0.1516 | 0.1717 | 0.2034 | 0.2789 |

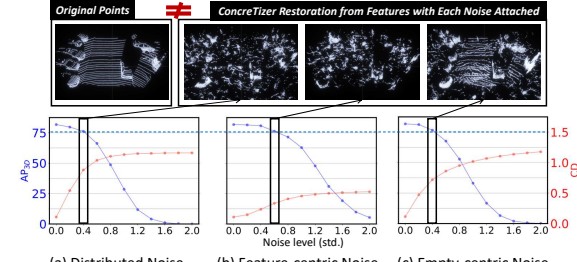

(a) Distributed Noise    (b) Feature-centric Noise    (c) Empty-centric Noise

Figure 9: **Effect of Gaussian noise.** Measured: $AP_{3D}$ (detection accuracy) of the SECOND model and CD (restoration error) of *ConcreTizer*.

As shown in Table 3 and Figure 9, these perturbations effectively reduce the restoration capability of the attack (defense) but also degrade object detection performance (target task), highlighting the challenge of mitigating *ConcreTizer* attacks without significantly compromising system utility. Notably, Figure 9 reveals that the sparse nature of 3D feature data causes noise to affect different regions unevenly, emphasizing the importance of considering spatial characteristics when designing future defense mechanisms. More visualization results are provided in the supplementary materials.

# 6 DISCUSSION: LIMITATIONS AND FUTURE DEFENSE STRATEGIES

Our inversion attack method demonstrates that 3D features are not inherently secure, as they can be exploited to restore the original data. This restored data could reveal private information, including personal identities, behavioral patterns, and location details, thereby posing a risk to the applications of voxel-based 3D vision models. Since *ConcreTizer* assumes that the parameters of the feature extractor are known, protecting model parameters can prevent such attacks. If sharing parameters is unavoidable, defense can be achieved by sacrificing some utility (accuracy of the vision model), as shown in Section 5.6. However, in accuracy-critical environments like autonomous driving, simple defense techniques are likely to be inadequate. Future research should focus on developing defenses that mitigate attacks while minimizing the impact on utility. Additionally, for latency-sensitive systems, it is crucial that defenses do not impose significant computational overhead. Potential strategies include Differential Privacy (DP) (Abadi et al., 2016), Adversarial Training (Liu et al., 2019), and Feature Obfuscation (Zhang et al., 2022). The strengths and limitations of each technique are discussed in the supplementary material.

# 7 CONCLUSION

This paper presents the first comprehensive study on model inversion for 3D point cloud restoration. In the context of autonomous driving, we focus on the most dominant voxel-based feature extractors and examine the challenges arising from their interaction with 3D point cloud characteristics. Based on this, we introduce *ConcreTizer*, a simple yet effective inversion technique tailored for restoring 3D point data from features, which incorporates Voxel Occupancy Classification and Dispersion-Controlled Supervision. Through rigorous evaluations using prominent open-source datasets such as KITTI and Waymo, along with representative 3D feature extractors, we not only demonstrate the superiority of *ConcreTizer* but also analyze each of its components in detail for valuable insights. Our research reveals the vulnerability of 3D point cloud data to inversion attacks, emphasizing the urgent need to devise extensive defense strategies. While this work focuses on voxel-based representations, we see inversions attacks for more diverse representations of 3D data, such as point set and graph, as valuable future work.

## ACKNOWLEDGMENTS

This work was supported in part by Institute of Information & communications Technology Planning & Evaluation (IITP) grant funded by the Korea government (MSIT) (No. IITP-2025-2021-0-02048 & IITP-2025-RS-2024-00418784) and in part by the National Research Foundation (NRF) of Korea grant funded by the Korea government (MSIT) (No. RS-2023-00212780).

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
