# OpenReview forum: "ConcreTizer: Model Inversion Attack via Occupancy Classification and Dispersion Control for 3D Point Cloud Restoration"
_ICLR.cc/2025/Conference — ICLR 2025 Poster_

### Official Review · Reviewer_8SUr · 2024-10-23

**Soundness:** 3
**Presentation:** 3
**Contribution:** 3
**Rating:** 8
**Confidence:** 4

**Summary:**

The paper highlights the growing privacy concerns surrounding the use of 3D point cloud data in autonomous vehicles due to the sensitive information it contains. While model inversion attacks have been well-studied in 2D data contexts, their application to 3D point clouds has largely been unexplored. To address this, the authors present an in-depth study of model inversion attacks aimed at reconstructing 3D point cloud scenes. They identify challenges such as the inherent sparsity of 3D data and the ambiguity between empty and non-empty voxels after voxelization, which is worsened by voxel dispersion across feature extractor layers. To tackle these issues, they introduce ConcreTizer, a model inversion attack specifically designed for 3D point cloud data, incorporating Voxel Occupancy Classification and Dispersion-Controlled Supervision. Experiments on popular 3D feature extractors and datasets like KITTI and Waymo demonstrate ConcreTizer's effectiveness in restoring disrupted 3D point cloud scenes, underscoring the vulnerability of 3D data to such attacks and the urgent need for robust defense strategies.

**Strengths:**

1.Good writing. This paper is well-written and easy to understand for even outliers like me, since I am not an expert in the research domain of model inversion attacks.

2.Nice figure presentations. The nice figures including Figures 1-3 make the paper more clearly and visually pleasing.

3.Well-motivated scheme. The authors have justified the motivation and key intuition behind proposed research problem and solution.

**Weaknesses:**

1.More analysis about the failures of existing inversion attacks are needed. In the Introduction section, Lines 71-72, the authors just demonstrated that conventional attacks failed, however, the authors did not provide more insights behind this phenomenon. A detailed explanation will help us get a better understanding of the reason why the proposed attack is effective.

2.Limited scope of proposed attacks. The authors said that, "we delve into the phenomena that arise when the voxel-based feature extractors handle 3D point cloud scenes", however, not all point cloud networks need to realize voxelization during feature extracting [1]. Therefore, it seems that the first challenge cannot represent the majority point cloud inversion attack scenes, limiting the effectiveness scope of the proposed attack scheme.

3.I encourage the authors to conduct more experiments to make results more convincing, since the page limit is 10 in ICLR 2025, providing enough space to demonstrate more results. For instance, the authors are supposed to employ more 3D point cloud datasets (which may not only focus autonomous driving datasets)

4.More defense baselines are supposed to be used for evaluation. Many recent works about point cloud adversarial attacks [2-4] have  employed  popular defense schemes like random 3D data augmentations, SOR, and SRS to evaluate the point cloud attack scheme's robustness. Thus, I will appreciate it if the authors can provide more results about defense evaluations.

[1] Dynamic graph cnn for learning on point clouds. TOG 2019

[2] PointAPA: Towards availability poisoning attacks in 3D point clouds. ESORICS 2024

[3] PointCRT: Detecting backdoor in 3D point cloud via corruption robustness. ACM-MM 2023

[4] PointBA: Towards backdoor attacks in 3D point cloud. ICCV 2021

**Questions:**

As a minor suggestion, the authors could revise the statement in Line 47 to make this claim “However, efforts to protect raw data have primarily focused on 2D domains, rather than 3D data.“ more accurate. Since recently there is already some parallel studies that mainly focus on protecting 3D raw data privacy against unauthorized training [5], the authors can revise this sentence to suggest the difference between existing 3D works and this paper's concentration view and the importance of studying model inversion attacks in 3D point clouds.

[5] Unlearnable 3D Point Clouds: Class-wise Transformation Is All You Need. NeurIPS 2024

---

> ### Author Response · Authors · 2024-11-21
>
> **W1. More analysis about the failures of existing inversion attacks**
>
> We appreciate the reviewer's feedback. However,we carefully analyzed the phenomena observed when applying existing attack methods to design our proposed techniques (Section 3).
>
> First, we observed that the issue, where the number of restored points is significantly higher and the points tend to concentrate near the origin, arises from the semantic ambiguity of zero-padded voxels. To address this, we proposed VOC (Lines 73-80 & Section 4.3.1).
>
> Next, we found that the increased severity of this issue in deeper layers was due to VoI dispersion during the downsampling process, and we introduced the DCS method to prevent VoI dispersion (Lines 81-88 & Section 4.3.2).
>
> In the evaluation, we aimed to demonstrate the effectiveness of VOC and DCS through an ablation study (Section 5.4).
>
> ****
> **W2. Limited scope of proposed attacks to voxel-based feature extractors**
>
> Thank you for your valuable comment.
>
> As you pointed out, it is true that our inversion attack targets a specific feature extractor. However, despite this, our study remains meaningful for two key reasons.
>
> First, due to the nature of the inversion attack task, which aims to find the inverse process of a given feature extractor, an attack targeting voxel-based feature extractors is still significant. In other words, even though a particular model is targeted, if it is a model that is actually in use, the privacy leakage threat must be addressed.
>
> Second, point-based and voxel-based feature extractors are the two main approaches for processing point cloud data. While point-based models excel in capturing details in complex indoor scenes, their computational inefficiency with large point clouds makes them less suitable for real-time applications. In contrast, voxel-based models prioritize efficiency, making them the dominant choice in autonomous driving [1]. Therefore, this study is still meaningful, as it focuses on feature extractors that dominate the autonomous driving domain.
>
> [1] R. Qian, X. Lai, and X. Li, "3D object detection for autonomous driving: A survey," Pattern Recognition, vol. 130, p. 108796, 2022.
>
> ****
> **W3. More experiments (e.g., not only focus autonomous driving datasets)**
>
> We appreciate your feedback. However, the reason for 9 pages is due to the CFP recommendations.
> Moreover, we believe focusing solely on autonomous driving data is still meaningful, as many important studies have been conducted using only autonomous driving dataset [1-7].
> Therefore, we also focused our research on the representative autonomous driving dataset, and additional experimental results can be found in the supplementary materials.
>
> [1] Y. Lu et al., "An extensible framework for open heterogeneous collaborative perception," ICLR, 2024.
>
> [2] L. Zhang et al., "Learning unsupervised world models for autonomous driving via discrete diffusion," ICLR, 2024.
>
> [3] S. Zhou et al., "Lidar-ptq: Post-training quantization for point cloud 3d object detection," ICLR, 2024.
>
> [4] Y. Yang, L. Fan, and Z. Zhang, "Mixsup: Mixed-grained supervision for label-efficient lidar-based 3d object detection," ICLR, 2024.
>
> [5] B. Zhang et al., "Resimad: Zero-shot 3d domain transfer for autonomous driving with source reconstruction and target simulation," ICLR, 2024.
>
> [6] J. Nie et al., "Towards category unification of 3D single object tracking on point clouds," ICLR, 2024.
>
> [7] K. Cheng et al., "UC-NeRF: Neural Radiance Field for Under-Calibrated multi-view cameras in autonomous driving," ICLR, 2024.

---

> ### Author Response · Authors · 2024-11-21
>
> **W4. More defense baselines**
>
> Thank you for your valuable comment.
>
> First, the defense techniques presented focus on adversarial attacks, which are entirely different from inversion attacks. While adversarial attacks involve manipulating the dataset to disrupt the model's training or inference process, inversion attacks aim to uncover the model's reverse process to restore the original data.
>
> Defense methods for inversion attacks include differential privacy and adversarial training.
> Differential privacy ensures privacy by adding Gaussian or Laplacian noise according to a mathematically defined privacy budget. [1,2] However, as shown in our experiments, the approach of sacrificing utility to protect privacy is unsuitable for applications like autonomous driving, where utility is critical.
> Adversarial training strengthens robustness against a specific attack model by adversarially training with the attack model. For example, in [3,4], generative models were used to create obfuscation, but this approach introduces significant overhead in terms of latency, making it difficult to integrate into autonomous driving applications.
>
> We have added a section (F.5) in the supplementary materials that discusses the potential defense mechanisms.
> However, please note that the primary contribution of this study is the first demonstration of the feasibility of inversion attacks. We expect future research to explore various defense techniques, with both attack and defense progressing in complementarity.
>
> [1] M. Abadi et al., "Deep learning with differential privacy," ACM SIGSAC, 2016.
>
> [2] Y. Zhao and J. Chen, "A survey on differential privacy for unstructured data content," ACM Computing Surveys, vol. 54, no. 10s, pp. 1-28, 2022.
>
> [3] N. Raval, A. Machanavajjhala, and J. Pan. "Olympus: Sensor privacy through utility aware obfuscation," Privacy Enhancing Technologies, 2019.
>
> [4] Z. Wu et al., "Towards privacy-preserving visual recognition via adversarial training: A pilot study," ECCV, 2018.
>
> ****
> **Q1. Writing revision “However, efforts to protect raw data have primarily focused on 2D domains, rather than 3D data.”**
>
> Thank you for your suggestion.
>
> First of all, [5] focuses on the "authority" of 3D point cloud data, which differs from our inversion attacks that address privacy leakage from the 3D shape information itself.
> What we intended to convey is that while inversion attack research is actively being pursued in the 2D domain, there has been limited research on this topic in the 3D point cloud domain. A more refined way to phrase this could be:
> *"However, research on privacy in 3D point cloud data remains limited and has not been explored as extensively as in the 2D image domain."*

---

> ### Comment · Reviewer_8SUr · 2024-11-22
> **Thanks for your rebuttal.**
>
> For W1: I still cannot find the insights of why existing inversion attacks failed. I suggest the authors give more analysis about this
> phenomena in the intro. The response seems to present "hypothesis" and "assumption" rather than analysis.
>
> For W2: I understand and acknowledge the significance of studying voxel-based feature extractors. While as the authors have realized that the scope is indeed limited, I would like to see more results of evaluation of point-based feature extractors. Besides, I would suggest the authors to add some intuitions about why this research only focuses on voxel-based feature extractors, which is beneficial for readers to understand the scope of this paper. Therefore, this rebuttal does not directly respond my concern.
>
> For W3: Thanks for your kind remind. This concern has largely been solved. I have a question about your recommended references: "Lidar-ptq: Post-training quantization for point cloud 3d object detection", where I wonder what is the difference between 3D object detection dataset and 3D point cloud autonomous driving data. Besides, I still believe that adding more results of 3D point cloud data is helpful to improving the paper quality.
>
> For W4: I am sorry, because I respectfully disagree with your claim that since random augmentations are designed for defending against adversarial attacks, they cannot be used to defend against inversion attacks. Adversarial training is also designed for adversarial attacks, however, it is employed to defend against backdoor attacks, poisoning attacks, and inversion attacks as you indicate. Besides, the popular defenses random augmentations, SOR, SRS, are data-centered defenses, which can be easily applied when defending against these attacks. Their aims are not limited to any scope of attacks. Random augmentations can also be used to defend against 3D point cloud backdoor attacks, poisoning attacks, and so on. Hence, this concern still remains.
>
> In conclusion, my concerns remain, I would like to encourage authors to provide more convincing results.

---

> ### Author Response · Authors · 2024-11-24
> **Follow-up for W3 and W4**
>
> **W3. ‘3D object detection dataset’ and ‘3D point cloud autonomous driving data’**
>
> Thank you for the question.
>
> In computer vision, there are various types of datasets and tasks. One way to represent 3D geometric information is through 3D point cloud data, which consists of a set of 3D coordinates. This data can be classified into three main categories based on the scope of the dataset:
>
> * Object-level datasets: These include datasets like [1], which are used for tasks focused on individual objects.
>
> * Indoor datasets: These are collected from indoor environments and are typically used for indoor scene analysis [2], [3].
>
> * Large-scale outdoor datasets: These datasets are collected from outdoor environments, such as those used in autonomous vehicles [4], [5]. They are typically large and sparse, which is why **voxel-based feature extractors** are commonly used for processing such data, **significantly outperforming point-based extractors**.
>
> The main tasks related to 3D point cloud data are classification, segmentation, and object detection. Classification models are primarily used for object-level datasets, while object detection models are used for scene-level datasets.
> Our research focuses on **large-scale outdoor datasets for autonomous vehicles**, which is also a 3D object detection dataset.
>
> [1] A. X. Chang et al., "Shapenet: An information-rich 3d model repository," arXiv preprint, 2015.
> [2] S. Song et al., "Sun rgb-d: A rgb-d scene understanding benchmark suite," IEEE CVPR, 2015.
> [3] A. Dai et al., "Scannet: Richly-annotated 3d reconstructions of indoor scenes," IEEE CVPR, 2017.
> [4] A. Geiger, P. Lenz, and R. Urtasun, "Are we ready for autonomous driving? the kitti vision benchmark suite," IEEE CVPR, 2012.
> [5] P. Sun et al., "Scalability in perception for autonomous driving: Waymo open dataset," IEEE/CVF CVPR, 2020.
>
> ---
> **W4. More defense baselines**
>
> We apologize for the misunderstanding and thank you for the detailed explanation.
>
> As you suggested, we tested the rotation, scaling, and random sampling methods on our inversion attack scenario. Below are the setups we used:
>
> * Rotation: Randomly selected from the range of -a to +a degrees (a = 1,2,3,4,5).
>
> * Scaling: Randomly selected from the range of -s to +s percent (s = 2.5,5,7.5,10).
>
> * Random Sampling: Randomly select r% of points (r = 80,60,40, 20,15,10,5).
>
> | Rotation (°) | 0       | 1       | 2       | 3       | 4       | 5       |
> |--------------|---------|---------|---------|---------|---------|---------|
> | **AP**       | 81.7702 | 38.3764 | 17.0799 | 12.0967 | 6.1025  | 2.7797  |
> | **CD**       | 0.0776  | 0.1142  | 0.1728  | 0.2310  | 0.2848  | 0.3344  |
>
>
> | Scaling (%)  | 0       | 2.5     | 5       | 7.5     | 10      |
> |--------------|---------|---------|---------|---------|---------|
> | **AP**       | 81.7702 | 44.2184 | 17.4126 | 9.6410  | 5.2598  |
> | **CD**       | 0.0776  | 0.1690  | 0.2542  | 0.3086  | 0.3516  |
>
>
> | Sampling (%) | 100     | 80      | 60      | 40      | 20      | 15      | 10      | 5       |
> |--------------|---------|---------|---------|---------|---------|---------|---------|---------|
> | **AP**       | 81.7702 | 77.8460 | 77.2418 | 72.5357 | 58.3154 | 52.7130 | 40.3081 | 24.5862 |
> | **CD**       | 0.0776  | 0.0794  | 0.0891  | 0.1080  | 0.1516  | 0.1717  | 0.2034  | 0.2789  |
>
> Similar to the Gaussian Noise method used in the original manuscript, we confirmed that all the three defense methods have the trade-off between utility (Average precision for 3D object detection accuracy) and privacy protection (chamfer distance for point cloud similarity). Experimental results showed that as the intensity of augmentation increased, the drop in AP became significantly larger across all methods. **No method is free from the trade-off between privacy and utility**, which highlights the challenges of defending against inversion attacks in applications like autonomous driving, which demand high accuracy.

---

> ### Author Response · Authors · 2024-11-25
> **Follow-up for W1**
>
> **W1. Regarding the insights of why existing inversion attacks failed**
>
> Sorry for using confusing words in the first response. Our insights come from neither hypothesis nor assumption, but from thorough analysis of voxel-based 3D feature extractors. We have carefully analyzed restoration patterns when applying existing methods to voxel-based feature extractors, identifying the following limitations of existing inversion attack methods:
>
>  - **[Observation 1] The number of restored points significantly exceeded the original data**: We identified an issue in voxel-based models related to the voxelization process. During voxelization, regions without points are zero-padded. However, conventional regression-based restoration methods do not consider point existence but focus solely on point localization, mistakenly interpreting zero-padded representations as valid points located at (0, 0, 0). As a result, points are created even for voxels that are actually empty, leading to an overgeneration of points compared to the original data.
>
> - **[Observation 2] Restored points were densely concentrated near the origin**: The inherently sparse nature of point clouds results in a large number of zero-padded voxels, far exceeding those containing valid points. Point regression-based inversion attacks naturally prioritize minimizing estimation errors across all voxels, including these zero-padded regions. Consequently, this bias towards zero-padded voxels causes an over-concentration of points near the origin in the restored scene. Moreover, it significantly increases localization errors for the relatively smaller number of valid points, as these errors are considered negligible within the overall regression error.
>
> - **[Solution 1] VOC**: To address this, we designed the VOC method to eliminate the semantic ambiguity of zero-padded voxels and focus on identifying the existence of points within a voxel rather than their precise localization, effectively mitigating the issues observed.
>
>
> ---
> - **[Observation 3] The negative impact of zero-padded voxels increases with layer depth**: By adjusting the depth of the feature extractor, we observed that as the layers deepen, existing attack methods are increasingly hindered by two aforementioned  limitations: (1) an excessive number of restored points and (2) an intensified concentration of points near the origin. This indicates that the "negative impact" of zero-padded voxels becomes more pronounced with greater feature extractor depth. Specifically, if voxels with a value of (0, 0, 0) persist in the final restored state, they are excluded from the regression targets and do not directly affect the regression loss. However, due to the nature of convolution operations, the values of non-empty voxels—defined as Voxels of Interest (VoI)—gradually disperse into the surrounding empty voxels. As the layers deepen, more originally zero-padded voxels become non-empty during the feature extraction and inversion processes. Consequently, **an increasing number of these zero-padded voxels are included in the regression targets.** Our experiments revealed that **the density of VoI spikes significantly at downsampling layers**, further amplifying the influence of zero-padded voxels on the final restoration results. This explains why the negative impact of zero-padded voxels escalates with the depth of the feature extractor.
>
> - **[Solution 2] DCS**: To address this, we proposed the DCS method, which introduces supervision based on the downsampling layers to suppress VoI dispersion, and verified its effectiveness.
>
>
> In the introduction, we intuitively connected the issues in existing methods to our proposed solutions (VOC and DCS) (Lines 72-88). We then dedicated a separate section (i.e., Section 3) to provide a more detailed explanation of these identified problems and analyses. In addition, **we will add more details mentioned above and an additional figure to describe these issues.**

---

> ### Author Response · Authors · 2024-11-25
> **Follow-up for W2**
>
> **W2-1: Reason why solely focus on voxel-based feature extractors**
>
> - We focused on voxel-based feature extractors because they **dominate the autonomous driving domain**, which is our target application requiring the analysis of **large-scale outdoor scenes** [1]. For instance, among the Top-5 published models for 3D object detection on the widely recognized KITTI benchmark, four are based on voxel representations [2–4], one is representation-agnostic [5], and none use point-based extractors.
>
> - Taking a deeper look, voxel-based extractors process point clouds by converting them into grid-like voxels, which significantly **reduces computational complexity** when dealing with large-scale outdoor scenes containing an extremely high number of points. Furthermore, for sparse data, these models leverage **specialized sparse convolution** techniques to enhance computational efficiency. This combination of scalability and efficiency makes voxel-based models particularly well-suited for real-time applications like autonomous driving.
>
> - These points are highlighted in Section 2 (Lines 126–129) and Section 4.1 (Lines 204–208) of the main text. To address the scope limitation, we will also **clarify this focus in the conclusion.**
>
> ---
> **W2-2: -Evaluation with point-based feature extractors**
>
> - Voxel-based and point set-based representations are two of the most widely used formats for 3D point cloud data, each with distinct characteristics, advantages, and limitations, **typically tailored to different applications**. Consequently, research has focused on developing feature extractors specialized for each representation rather than unified models, resulting in significant architectural differences between the two.
>
> - Specifically, in voxel-based models, zero-padding-based representations for empty regions introduce challenges, and downsampling via convolution causes the dispersion of the Voxels of Interest (VoI) into neighboring regions. Our VOC and DCS methods were specifically designed to address these issues. In contrast, point-based and graph-based models exclusively handle valid points and employ different techniques like Farthest Point Sampling (FPS) for downsampling, resulting in increasingly sparse point clouds. Consequently, **challenges arising from the interaction of empty and non-empty regions in voxel-based models are absent in these architectures. Instead, addressing sparsity through effective point upsampling techniques becomes critical (e.g., [6]).** While the progressive restoration concept in DCS has the potential for broader applicability across different data representations, we believe that point- and graph-based models require **different attack methods** tailored to their unique characteristics.
>
>
> - Given these differences, creating a single inversion attack model that works across both architectural types is not only challenging but also inefficient. It is more practical to design inversion attacks tailored to the unique characteristics of each representation. **Since this paper targets autonomous driving applications, which predominantly use voxel-based models, evaluating point set-based models is outside the scope of this work, needs another separately designed method, and would add limited value.** However, developing an inversion attack for point set-based models could be a valuable direction for future work, particularly for applications focused on indoor scene analysis and smaller-scale objects.
>
> [1] R. Qian, X. Lai, and X. Li, "3D object detection for autonomous driving: A survey," Pattern Recognition, vol. 130, p. 108796, 2022.
>
> [2] H. Wu et al., "Virtual sparse convolution for multimodal 3d object detection," IEEE/CVF CVPR, 2023.
>
> [3] P. Gao and P. Zhang, "MPCF: Multi-Phase Consolidated Fusion for Multi-Modal 3D Object Detection with Pseudo Point Cloud," 2024.
>
> [4] H. Hoang, D. Bui, and M. Yoo, "TSSTDet: Transformation-Based 3-D Object Detection via a Spatial Shape Transformer," IEEE Sensors Journal, 2024.
>
> [5] Z. Dong et al., "PeP: a Point enhanced Painting method for unified point cloud tasks," arXiv preprint, 2023.
>
> [6] Y. He et al., "Grad-pu: Arbitrary-scale point cloud upsampling via gradient descent with learned distance functions," IEEE/CVF CVPR, 2023.

---

> ### Comment · Reviewer_8SUr · 2024-11-25
> **Thanks for the authors' effort.**
>
> For W1: The authors said, "We have carefully analyzed restoration patterns when applying existing methods to voxel-based feature extractors", however, I cannot grasp what the "existing method" is, and the authors even do not cite or display this method to show me intuitively. After reading this rebuttal, I still don't know what existing inversion attack the author used to reveal the relevant analysis, which makes me less convincing about the authors' rebuttal. I hope the authors can directly and clearly respond to my concerns in future rebuttals, which will help me quickly grasp the key points.
>
> For W2-1: This response is better than previous one. I am satisfied with this response. It is suggested that the authors add relevant statement like "We focused on voxel-based feature extractors because they dominate the autonomous driving domain" in the main text.
>
> For W2-2: It is obvious that the scheme designed for voxel-based data is less effective against point-based data. However, it is important to present some experimental results to discuss this point, which will make this paper more complete. Therefore, this response still does not respond my concern.
>
> For W3: This response about the taxonomy of point cloud datasets seems messy. Indoor datasets and large-scale outdoor datasets also include object-level datasets, and the authors said, "object detection models are used for scene-level datasets", I do not see the scene-level datasets in the taxonomy list. Why is it? By the way, object-level dataset can also be used to conduct object detection tasks, which the authors may have ignored.
>
> For W4: It is strange that the author used a rotation of 1 ° but caused such a significant change in indicators. I think this is unusual. Can the authors explain this? Similarly, for scaling of 2.5%, the results also seem to be unusual.

---

> ### Author Response · Authors · 2024-11-25
> **Further efforts to address W1 and W3**
>
> **W2-1. About scope of proposed attacks**
>
> Thank you for understanding. We will include relevant statements in the main text.
>
> ---
> **W1. About ‘existing method’**
>
> We employed the approach described in [1] and referred to it as **point regression**. As mentioned in Lines 50–53 of the main manuscript, this is **the only relevant (but ultimately unsuccessful) prior method**, as our work is the first to deeply investigate this regime.
>
> Similar to ConcreTizer, point regression utilizes a symmetric model structure to restore the original data shape from downsampled features by inverting a target feature extractor. Specifically, the convolution layers used for downsampling are mirrored by corresponding transposed convolution layers for upsampling. The final output matches the spatial dimensions of the original voxelized data, with a channel size of 3, where the channel values represent the 3D location coordinates (x, y, z).
>
> However, point regression differs from ConcreTizer in the following key ways, leading to the limitations outlined in our earlier rebuttal:
>
> - **Localization loss:** Point regression focuses on point localization by directly estimating each voxel's channel values. The generated output of the point regression model is trained by calculating the mean squared error (MSE) of the channel values at each voxel compared to the original data. The localization task is harder and more error-prone than classification.
>
> - **Naive determination of point existence:** When a voxel's regression output is exactly (0, 0, 0), it indicates that the zero-padded voxel remains unaffected by the entire forward and inversion process. Such voxels are treated as empty regions and excluded from the MSE calculation. If a voxel output deviates from (0, 0, 0), it is included in the MSE calculation (defined as a Voxel of Interest (VoI) in our paper) and generates a valid point in the restored scene.
>
> - **Problems (summarized from the previous rebuttal):** As VoIs disperse into empty voxels during the forward and inversion processes—especially in deeper feature extractors—point regression falsely generates points for a significant number of originally zero-padded (empty) voxels. Moreover, these numerous false VoIs (originally zero-padded) disproportionately influence the MSE loss, driving the point regression model to generate most points near the origin (0, 0, 0) to minimize estimation error for these false VoIs. This bias severely impacts localization performance for the relatively smaller number of true VoIs (originally non-empty voxels).
>
> We hope that this detailed explanation of the point regression method, along with the observations on its limitations provided in our previous response, addresses the reviewer’s concerns.
>
> ---
> **W3. Taxonomy of point cloud datasets**
>
> To refine the taxonomy, point cloud data can be broadly categorized into object-level and scene-level datasets. Scene-level datasets can be further divided into indoor and outdoor data.
>
> - Regarding your comment about "conducting object detection on object-level datasets," it is important to note a key distinction between 2D images and 3D point cloud data. In 2D images, even for object-level datasets, background information is often included, making object detection a meaningful task. However, in 3D point cloud datasets at the object level (e.g., [2]), **background information is absent**, and the data typically represents only a single object, such as a car, chair, or airplane. In such cases, **object detection is irrelevant**, and these datasets are not used for object detection tasks.
>
> - In contrast, scene-level 3D point cloud datasets include both object and background information, making object detection tasks meaningful and applicable. Within scene-level datasets, indoor datasets (e.g., [3, 4]) generally feature relatively evenly distributed points within confined spaces. **Outdoor datasets (e.g., [5, 6]), on the other hand, cover large-scale environments and exhibit varying point densities depending on the distance from the sensor.**
>
> [1] S. Hwang et al., "UpCycling: Semi-supervised 3D Object Detection without Sharing Raw-level Unlabeled Scenes," IEEE/CVF ICCV, 2023.
>
> [2] A. X. Chang et al., "Shapenet: An information-rich 3d model repository," arXiv preprint, 2015.
>
> [3] S. Song et al., "Sun rgb-d: A rgb-d scene understanding benchmark suite," IEEE CVPR, 2015.
>
> [4] A. Dai et al., "Scannet: Richly-annotated 3d reconstructions of indoor scenes," IEEE CVPR, 2017.
>
> [5] A. Geiger, P. Lenz, and R. Urtasun, "Are we ready for autonomous driving? the kitti vision benchmark suite," IEEE CVPR, 2012.
>
> [6] P. Sun et al., "Scalability in perception for autonomous driving: Waymo open dataset," IEEE/CVF CVPR, 2020.

---

> > ### Author Response · Authors · 2024-11-25
> > **Further efforts to address W2-2 and W4**
> >
> > **W2-2. Evaluation with point-based feature extractors**
> >
> > We respectfully disagree with the reviewer’s statement: “It is obvious that the scheme designed for voxel-based data is less effective against point-based data.” **The issue is not one of effectiveness but of feasibility.** Voxel-based models requires a point cloud to be voxelized before processing, making it impractical to directly handle original point set-based representations. Accordingly, to the best of our knowledge, **no existing work** has attempted to forcibly evaluate a voxel-based model directly on point set-based representations.
> >
> > As we have clarified, our research specifically targets voxel-based models, and ConcreTizer cannot be directly applied to point set-based models. For instance, applying the VOC module to point set-based models, which do not utilize voxel representations, is **simply not feasible**.
> >
> > We view the development of an inversion attack method for point set-based models as an important direction for future work and kindly ask for the reviewer’s understanding regarding the reason of not including additional experiments.
> >
> > ---
> > **W4. Significant accuracy drop even with weak intensity augmentation**
> >
> > As discussed in W3, autonomous driving datasets consist of **large-scale, scene-level data**, where even augmentations with weak intensity can lead to significant distortions over long distances, potentially causing considerable performance degradation.
> >
> > For example, a small 1-degree rotation can result in up to 1.4 meters of positional distortion within the maximum detection range (80 meters) of the KITTI dataset. Similarly, a 2.5% scaling can introduce distortions of up to 2 meters. These distortions can severely impact the performance of 3D object detection tasks, where accurately determining object locations through 3D bounding boxes is critical.

---

> ### Comment · Reviewer_8SUr · 2024-11-27
> **Thanks for further clarifications.**
>
> For W1: My concern is consistently related to evaluation of existing inversion attacks' failures, which can be seen from my original comment "More analysis about the failures of existing inversion attacks are needed". However, the "existing method" [1] provided by the authors seems to a SSL training scheme for object detection, which makes me confused. This response is less convincing from my perspective.
>
> For W2-2: Thanks for the authors' clarifications. This response has addressed my concern. In conclusion, the inversion attack proposed by this paper is limited to voxel-based point cloud data. Therefore, I urge the authors to revise the overclaimed statements in the manuscript such as "we introduce ConcreTizer, a simple yet effective model inversion attack designed specifically for 3D point cloud data". It should be "designed specifically for voxel-based 3D point cloud data".
>
> For W3: A quick question: Why does the 3D object detection needs the background infromation?
>
> For W4: For the authors' results and explanations, it seems that very slight augmentations like rotation, scaling might degrade the object detection performance,  however, in 3D point cloud classification tasks, these random augmentations are beneficial for the performance [2]. Therefore, it is counter intuitive. Does it mean that we cannot train a 3D object detection model with data augmentation schemes? I think it is an interesting point to be discussed in the manuscript.
>
> From my side, I really appreciate the efforts that the authors put to address my concerns. I would like to see more clarifications to thoroughly address my questions. Thanks for the authors' effort again.
>
> [1] S. Hwang et al., "UpCycling: Semi-supervised 3D Object Detection without Sharing Raw-level Unlabeled Scenes," IEEE/CVF ICCV, 2023.
>
> [2] Wang, Xianlong, et al. "Unlearnable 3D Point Clouds: Class-wise Transformation Is All You Need." The Thirty-eighth Annual Conference on Neural Information Processing Systems (NeurIPS 2024).

---

> ### Author Response · Authors · 2024-11-28
> **Further efforts to address the concerns**
>
> We appreciate the reviewer's response and have made every effort to address the concerns as detailed below:
>
> ---
> **W1. "existing method" [1] seems to a SSL training scheme**
>
> We would like to clarify our stance on this matter. As emphasized in the main manuscript, we are **the first to explore inversion attacks on voxel-based 3D feature extractors.** Strictly speaking, no established methods currently exist for this specific task.
>
> - To the best of our knowledge, the work in [1] is the only prior study mentioning an inversion attack for 3D point cloud data. However, as the reviewer pointed out, even this work does not focus on inversion attacks. Instead, it aims to leverage unlabeled 3D features under the assumption that they are inherently safe and free from privacy concerns. **In the absence of dedicated inversion attack methods for 3D data, the authors of [1] developed a point regression-based attack method to test their hypothesis.** Their findings indicated that this approach failed to restore original scenes from extracted features, which they presented as evidence supporting the inherent privacy-preserving nature of 3D features.
>
> - However, we disagree with this conclusion. We argue that the failure of their method is **not due to an inherent safety of 3D features but rather a lack of careful design that accounts for the unique characteristics of 3D backbones.** To substantiate this claim, we analyzed the operation of point regression on voxel-based feature extractors, identifying key limitations of the approach.
>
> These details are now included in Sections 1 and 3 of the revised manuscript, and we hope that this can address the reviewer’s concern.
>
> ---
> **W2-2. Evaluation with point-based feature extractors**
>
> We are pleased to hear that your concerns have been addressed. Also, thank you for the suggestion! In response, we have revised the relevant statements in the abstract, introduction, and conclusion, and have uploaded the updated manuscript.
>
> ---
> **W3. Why does the 3D object detection need the background information?**
>
> In short, **without the background information, there is no challenge to “detect” an object, as there is nothing further to distinguish.**
> - Specifically, 3D object detection involves estimating the positions, sizes, and orientations of objects within a scene, typically using 3D bounding boxes. The main goal is to identify and separate individual objects within a complex environment that contains multiple objects and background elements.
> - However, in object-level datasets, which lack background information, the objects are **already clearly defined and separated.** This eliminates the need to distinguish object-related points (foreground) from background points. Instead, research in such datasets typically shifts focus to analyzing the shape and characteristics of the objects.
>
> ---
> **W4. Significant accuracy drop even with weak intensity augmentation**
>
> We understand the reviewer’s question but would like to clarify that **this phenomenon is not counter-intuitive but a well-known aspect of 3D data processing.** The key distinction lies in how augmentation techniques, such as scaling and rotation, are **applied differently** in 3D classification and 3D object detection tasks:
>
> - **For classification tasks,** the label corresponds to the entire point cloud and represents the object’s category. Transformations like rotation or scaling do not alter the label, as they preserve the object’s overall shape and category. Thus, these augmentations can enhance the models’ robustness and improve performance.
>
> - **For object detection tasks,** the labels include not only the object category but also **precise location details** such as position, size, and orientation within a complex scene. When transformations like rotation or scaling are applied, the ground-truth location information in the labels **must also be updated** to reflect these changes. During training, this adjustment is feasible since the labels are available, enabling effective data augmentation.
>
>
> However, during test-time defenses against inversion attacks, label information is unavailable, and data perturbations occur without corresponding label updates. **This mismatch between transformed data and unchanged labels leads to significant performance degradation.** This effect is particularly pronounced for distant objects, where small augmentations like rotation and scaling can result in large distortions. In contrast, random sampling does not require label adjustments, as it does not alter the location-based information in the labels. Consequently, its impact on performance is relatively minor.
>
>
> Ultimately, the experiments demonstrate a fundamental trade-off: achieving privacy through defenses against ConcreTizer’s inversion attacks comes at the cost of reduced utility. This trade-off underscores the challenges in defending against our attack methodology.

---

> > ### Comment · Reviewer_8SUr · 2024-11-28
> > **Thanks for the authors' efforts.**
> >
> > Thanks for the authors' detailed response. After carefully reading these responses, most of my concerns have been well addressed. I believe that this paper has its merit in the domain of adversarial machine learning on 3D point cloud data.
> >
> > In addition, from the multiple rounds of communication with the authors, I find that the biggest issues of this original submission are
> > - Overclaims about the effective scope of the proposed attack;
> > - Unclear descriptions of the evaluation of autonomous datasets and object detection datasets;
> > - Insufficient discussion about relevant works (including possible defense schemes like random rotation and scaling [1-2, 4], relevant analysis of existing attacks [3], and related works about 3D point cloud data privacy [4]).
> >
> > Despite these issues accompanied by the initial submission, after the rebuttals, I believe that the authors are able to well address these issues I mentioned and I would like increase the score of this paper once the authors revised their paper accordingly.
> >
> > Thanks for the authors' dedicated efforts during rebuttal period again.
> >
> > [1] PointAPA: Towards availability poisoning attacks in 3D point clouds. ESORICS 2024
> >
> > [2] PointBA: Towards backdoor attacks in 3D point cloud. ICCV 2021
> >
> > [3] UpCycling: Semi-supervised 3D Object Detection without Sharing Raw-level Unlabeled Scenes. ICCV 2023
> >
> > [4] Unlearnable 3D Point Clouds: Class-wise Transformation Is All You Need. NeurIPS 2024

---

> ### Author Response · Authors · 2024-11-28
> **Thanks for the positive response.**
>
> Thank you for the reviewer’s positive feedback on our rebuttal and for considering raising the score.
>
> We deeply appreciate the productive discussion during the rebuttal process. We have carefully revised our manuscript to address the reviewer’s comments and incorporated the three additional papers [1, 2, 4] mentioned earlier. The latest version has been uploaded. While the rebuttal PDF deadline has passed, we welcome any further suggestions and would be glad to incorporate them to enhance the final manuscript if accepted.

---

> > ### Comment · Reviewer_8SUr · 2024-11-28
> > **I have updated my evaluation accordingly. All the best!**
> >
> > I appreciate the efforts that the authors have made during the rebuttal period, which has finally addressed my concerns. I have updated my score and confidence evaluation.

---

### Official Review · Reviewer_VptB · 2024-11-03

**Soundness:** 2
**Presentation:** 3
**Contribution:** 2
**Rating:** 5
**Confidence:** 4

**Summary:**

The paper proposed the first study of model inversion attacks for 3D point cloud data. The primary contributions of our work include (1) identifying the challenges associated with the interaction between 3D point cloud characteristics and voxel-based feature extractors, (2) introducting ConcreTizer that utilizes Voxel Occupancy Classification (VOC) and Dispersion-Controlled Supervision (DCS) to restore original 3D point clouds from their voxel-based features. (3) Experiments demonstrate the superior performance of ConcreTizer compared to a traditional method and GAN-based method. The findings highlight the vulnerability of 3D point clouds to inversion attacks and underscore the need for enhanced defensive strategies in privacy-sensitive applications.

**Strengths:**

1.	Originality: The paper introduces ConcreTizer, the first approach to 3D point cloud restoration from voxel-based features, highlighting previously unexamined vulnerabilities.
2.	Clarity: The paper is well-organized.
3.	Significance: By demonstrating the risks of model inversion attacks on voxel-based 3D features, this work has meaningful implications for privacy in applications like autonomous driving, emphasizing the need for robust defenses in this domain.

**Weaknesses:**

1.	Focus on Voxel-based Features Only:
The paper’s focus is solely on voxel-based feature inversion, with no consideration for point-based or projection-based features. Broadening to other feature types or discussing ConcreTizer’s potential for generalization would enhance applicability.
2.	Limited Comparison Methods:
ConcreTizer is mainly compared with Point Regression and UltraLiDAR. Including more inversion methods, such as recent GAN-based or diffusion-based approaches, would provide a more thorough evaluation and highlight ConcreTizer's strengths more effectively.
3.	Limited Validation Scope:
a)	3D Object Detection Evaluation: Only PointPillar is used to assess detection on restored scenes. Testing with other models (e.g., CenterPoint, PV-RCNN…) would provide better insight into generalizability.
b)	Effect of DCS Instances in Ablation Study: The chosen values for DCS instances (1, 2, 4, 10) are limited and unevenly spaced. Testing a broader, more evenly distributed range could offer clearer insights into optimal DCS configuration.
4.	Writing:
a)	Why the results for VoxelResBackbone with the Waymo dataset are not included?
b)	It’s unclear if "VOC alone" in the component-wise analysis ablation study refers solely to using the regression loss function. Explicit clarification would help.
c)	The phrase, “Since the restoration task does not require separate labeling, it is possible to train using open source datasets or self-collected data,” could be streamlined as the first part might appears redundant.

**Questions:**

Please refer to the weakness part.

---

> ### Author Response · Authors · 2024-11-21
> **Addressing W1**
>
> **W1. Focus on Voxel-based Features Only (Broadening to point-based or projection-based features)**
>
> Thank you for your valuable comment.
>
> **Rationale for focusing on voxel-based feature extractors**
>
> - The primary reason for focusing on voxel-based feature extractors is their **dominance in the autonomous driving domain**, which is our target application requiring the analysis of large-scale outdoor scenes [1]. For example, among the Top-5 published models for 3D object detection on the widely recognized KITTI benchmark, four use voxel representations [2–4], one is representation-agnostic [5], and **none employ point-based or projection-based extractors.**
>
> - Specifically, voxel-based extractors process point clouds by converting them into grid-like voxels, significantly reducing computational complexity when handling large-scale outdoor scenes with an extremely high number of points. Furthermore, these models employ specialized sparse convolution techniques to enhance efficiency when dealing with sparse data. This scalability and efficiency make voxel-based models highly suitable for real-time applications like autonomous driving.
>
> - In contrast, point-based methods struggle with high computational demands as point cloud sizes grow, making them less practical for real-time scenarios. Similarly, while projection-based methods are computationally simpler, they lose substantial 3D information, which limits their accuracy in complex scenes.
>
>
>
> **Applicability of ConcreTizer to other feature extractors (need for future work)**
>
> Voxel-based and point set-based representations are most widely used formats for 3D point cloud data, each tailored to different applications with distinct characteristics. Accordingly, research has focused on feature extractors specialized for each representation, leading to significant architectural differences between them. Given these differences, designing a single inversion attack that functions effectively across both architectures is not only challenging but also inefficient. It is more practical to create attacks tailored to the unique characteristics of each representation.
>
> - **Voxel-Based Models (our target):** In voxel-based models, zero-padding for empty regions introduces challenges, and downsampling through convolution disperses Voxels of Interest (VoI) into neighboring regions. Our VOC and DCS methods were specifically designed to address these issues.
>
> - **Point-Based Models:** Point-based models [6, 7] exclusively handle valid points and employ techniques like Farthest Point Sampling (FPS) for downsampling, resulting in increasingly sparse point clouds. Consequently, challenges related to interactions between empty and non-empty regions in voxel-based models do not exist in point-based architectures. Instead, addressing sparsity through effective point upsampling techniques becomes critical (e.g., [7]). While the progressive restoration concept of DCS could have broader applicability, point- and graph-based models would require tailored attack methods to account for their unique characteristics.
>
> - **Projection-Based Models:** In projection-based models [8, 9], 3D point cloud data is projected along a specific axis (e.g., the z-axis or radial axis) and converted into a 2D representation for processing with 2D convolutions. While the restoration of the 2D CNN component could leverage existing 2D inversion methods, the projection restoration process would likely pose additional challenges, requiring further development.
>
> We hope this explanation clarifies the rationale behind our focus on voxel-based models and highlights opportunities for future research in point-based and projection-based inversion attacks.
>
>
> [1] R. Qian, X. Lai, and X. Li, "3D object detection for autonomous driving: A survey," Pattern Recognition, vol. 130, p. 108796, 2022.
>
> [2] H. Wu et al., "Virtual sparse convolution for multimodal 3d object detection," IEEE/CVF CVPR, 2023.
>
> [3] P. Gao and P. Zhang, "MPCF: Multi-Phase Consolidated Fusion for Multi-Modal 3D Object Detection with Pseudo Point Cloud," 2024.
>
> [4] H. Hoang, D. Bui, and M. Yoo, "TSSTDet: Transformation-Based 3-D Object Detection via a Spatial Shape Transformer," IEEE Sensors Journal, 2024.
>
> [5] Z. Dong et al., "PeP: a Point enhanced Painting method for unified point cloud tasks," arXiv preprint, 2023.
>
> [6] C. R. Qi et al., "Pointnet++: Deep hierarchical feature learning on point sets in a metric space," NeurIPS, 2017.
>
> [7] Y. He et al., "Grad-pu: Arbitrary-scale point cloud upsampling via gradient descent with learned distance functions," IEEE/CVF CVPR, 2023.
>
> [8] B. Yang, W. Luo, and R. Urtasun, "Pixor: Real-time 3d object detection from point clouds," IEEE CVPR, 2018.
>
> [9] X. Chen et al., "Multi-view 3d object detection network for autonomous driving," IEEE CVPR, 2017.

---

> ### Author Response · Authors · 2024-11-21
>
> **W2. Limited Comparison Methods (Including recent GAN-based or diffusion-based approaches)**
>
> We agree that incorporating generative models, such as GANs or diffusion models, would provide a more comprehensive evaluation. In fact, **this is precisely why we selected UltraLiDAR for our experiments.**
>
> While numerous studies have explored GAN- or diffusion-based point cloud generation [1–5], the majority focus on small object-scale point clouds [6], making their direct application to large-scale outdoor scenes, such as those encountered in autonomous driving scenarios, particularly challenging. Among generative models developed for point cloud scenes [7–9], UltraLiDAR [9] stands out as the only method capable of conditioning on voxel-based features within the context of autonomous driving—a key requirement of our target application. Given the need for analyzing large-scale outdoor scenes in autonomous driving, UltraLiDAR was the most appropriate choice for comparison.
>
> As generative models for point cloud scenes continue to evolve, exploring their potential for inversion attacks in large-scale scenarios is an exciting direction for future work.
>
>
>
> [1] P. Achlioptas et al., "Learning representations and generative models for 3d point clouds," PMLR ICML, 2018.
>
> [2] D, Valsesia, G. Fracastoro, and E. Magli, "Learning localized generative models for 3d point clouds via graph convolution," ICLR, 2018.
>
> [3] D. W. Shu, S. W. Park, and J. Kwon, "3D point cloud generative adversarial network based on tree structured graph convolutions," IEEE/CVF ICCV, 2019.
>
> [4] G. Yang et al., "Pointflow: 3d point cloud generation with continuous normalizing flows," IEEE/CVF ICCV, 2019.
>
> [5] S. Luo and W. Hu, "Diffusion probabilistic models for 3d point cloud generation," IEEE/CVF CVPR, 2021.
>
> [6] A. X. Chang et al., "Shapenet: An information-rich 3d model repository," arXiv preprint, 2015.
>
> [7] L. Caccia et al., "Deep generative modeling of LiDAR data," IEEE/RSJ IROS, 2019.
>
> [8] V. Zyrianov, X. Zhu, and S. Wang, "Learning to generate realistic LiDAR point clouds," ECCV, 2022.
>
> [9] Y. Xiong et al., "Learning compact representations for LiDAR completion and generation," IEEE/CVF CVPR, 2023.
>
>
> ****
> **W3. Limited Validation Scope**
>
> **a) 3D Object Detection Evaluation (other models (e.g., CenterPoint, PV-RCNN…))**
>
> Thank you for your suggestion.
>
> As our inversion attack produces 3D point cloud data, we primarily used metrics like Chamfer Distance (CD), Hausdorff Distance (HD), and F1 score to evaluate the similarity of restored point clouds directly. Based on your feedback, we are conducting **additional experiments** using other 3D object detection models. We will report these results as soon as the experiments are completed. While the ongoing experiments will provide further validation, we anticipate that the performance trends will remain consistent across different object detection models.
>
>
> **b) Effect of DCS Instances in Ablation Study (a broader, more evenly distributed range)**
>
> We apologize for any confusion in the ablation study. Since the feature extractor contains four downsampling layers, the relevant instances for DCS with **even splitting** are 1, 2, and 4. Layer 10 was **optionally included** as a test case where splitting occurs across all layers, regardless of the downsampling layers.
>
> To address concerns about the optimality of the DCS configuration, we have included additional experimental results in the supplementary materials (Section F.3 and Figures 10 and 11) for your review. Additionally, we are running an experiment with an **uneven DCS setup for three instances**, such as a split configuration of (2,1,1) across the four downsampling layers. We hope that  these results can help further clarify the impact of DCS configurations on performance.
>
>
> ****
> **W4. Writing**
>
> **a) Results for VoxelResBackbone with the Waymo dataset**
>
> The results from experiments with the VoxelResBackbone and the Waymo dataset are included in the supplementary materials. We included a mention of this in the main text (Line 406).
>
>
> **b) Meaning of "VOC alone" in the component-wise analysis ablation study**
>
> Our proposed method consists of VOC and DCS. Section 5.4 is an ablation study to evaluate the contribution of each component, and thus 'VOC alone' refers to the case where only VOC is used without DCS. While detailed information about VOC is provided in Section 4.3.1, to briefly explain, we replace the regression loss used in previous methods with a classification loss.
>
> **c) Redundancy ("Since the restoration task does not require~ ")**
>
> Thank you for your suggestion.
>
> After receiving your comment, the previous sentence is revised as:
> *“In this scenario, an attacker with access to the same feature extractor **would need to prepare 3D point cloud data for training the inversion attack model.** Since the restoration task doesn't require separate labeling, training can be conducted using open-source datasets or self-collected data.”*

---

> > ### Author Response · Authors · 2024-11-28
> > **Further Reports on W3 and Kind Reminder**
> >
> > Dear reviewer,
> >
> > We are pleased to inform you that the additional experiments you requested for addressing W3 have been completed, the revised manuscript has been uploaded. Below, we summarize the progress:
> >
> > **W3. Limited Validation Scope**
> >
> > - Based on your feedback, we conducted evaluations using various 3D object detection models including **PointPillar, PVRCNN, CenterPoint, VoxelRCNN, and PointRCNN.** The experimental results confirmed that our proposed method, ConcreTizer, consistently achieves the best performance regardless of the object detection model. The results can be found in **Table 2 on page 9** of the updated main paper.
> >
> > - Additionally, we conducted experiments for the case of **DCS instance 3.** Partitioning was performed to include 1/2/1 downsampling layers in the three respective blocks. The experimental results showed that partitioning with consideration for the downsampling layers (DCS 2, 3, and 4) outperformed both the case with no DCS (DCS 1) and the case with excessive DCS (DCS 10). The results can be found in **Figure 8 on page 10** of the updated main paper.
> >
> >
> >
> > As the PDF upload deadline is approaching, we kindly remind you that we are awaiting your response. We sincerely appreciate your continued efforts and thank you again for your time and consideration.

---

> > > ### Author Response · Authors · 2024-12-03
> > > **Last Reminder**
> > >
> > > Dear reviewer,
> > >
> > > We believe that our rebuttal has addressed your concerns.
> > >
> > > As the discussion deadline is approaching, please put your response as soon as possible.
> > >
> > > Our discussion with Reviewer 8SUr during the rebuttal period might help your decision.
> > >
> > > We appreciate your efforts.

---

### Official Review · Reviewer_3vLL · 2024-11-03

**Soundness:** 3
**Presentation:** 2
**Contribution:** 2
**Rating:** 5
**Confidence:** 4

**Summary:**

This paper presents the first comprehensive study of model inversion attacks in 3D point cloud reconstruction tasks, introducing an attack method named ConcreTizer. ConcreTizer utilizes Voxel Occupancy Classification (VOC) to distinguish between zero-padded empty voxels and non-empty voxels containing valid points at the origin (0, 0, 0), along with Dispersion-Controlled Supervision (DCS) to enforce sparsity in the recovered 3D point cloud data. Extensive quantitative and qualitative experiments on open-source datasets, such as KITTI and Waymo, validate the effectiveness of this inversion attack and underscore the privacy vulnerabilities and the critical need for robust defenses in 3D point cloud data applications.

**Strengths:**

- ConcreTizer represents the first exploration of inversion attacks on 3D point cloud data, providing valuable insights and inspiration for further research.
- The paper demonstrates ConcreTizer’s effectiveness across various datasets and feature extractors, achieving impressive point cloud reconstruction results. The breadth and diversity of experiments enhance the credibility of the findings.

**Weaknesses:**

1. The attack scenario in this paper lacks practical significance. One of the core innovations proposed by the paper is the introduction of VOC to differentiate between empty voxels and those located at the (0,0,0), as they share the same representation. However, a simpler solution exists to address this issue, such as adding an "occupied" label when voxelizing the point cloud data, which would render the VOC setup unnecessary.
2. The paper contains some ambiguities in its expression, making it difficult to understand. See Question 1 for further details.

**Questions:**

There are ambiguities in the presentation. For instance, in Figure 2, the white areas, as suggested by the context, represent actual point cloud data, which decreases with increasing layer depth. However, the caption of Figure 2 states, “As the layer depth increases, the number of restored points increases rapidly, ...” which contradicts the visual representation and may confuse readers trying to align the illustration with the description.

---

> ### Author Response · Authors · 2024-11-21
>
> **W1. A simpler solution exists to address this issue, such as adding an "occupied" label when voxelizing the point cloud data, which would render the VOC setup unnecessary.**
>
> Thank you for your comment but there seems to be a **misunderstanding**.
>
> In the inversion attack scenario, the attacker cannot manipulate the feature extraction process, including voxelization. As shown in the "ConcreTizer Attacker Scenario" at the top of Figure 3 (now Figure 4 in the revised manuscript), the attacker is only tasked with finding the inverse process based on the given feature extractor, with no control over its design.
> This means the attacker cannot add an "occupied" label **during voxelization**. In our approach, VOC generates and uses the "occupied" label exclusively during training. During inference, only features without the "occupied" label are shared, and the original data must be reconstructed from these features.
> This ensures the effectiveness and value of our VOC setup in our inversion attack scenario.
>
> ****
> **W2 (Q1). Ambiguities in Figure 2.**
>
> Thank you for your thorough feedback.
>
> To clarify, the caption is correct, but there was some confusion in the figure (**now, Figure 3 in the revised manuscript**).
> In Figure 2, the original data contains 14,137 points, while the number of restored points for layers 3, 6, 9, and 12 are 107,032, 252,449, 603,206, and 692,946, respectively, showing a rapid increase. However, the large number of points in the images for layers 9 and 12 caused an error during the visualization process.
> We have now **uploaded the revised figure**. Please review it again.
> Thank you.

---

> > ### Author Response · Authors · 2024-11-28
> > **Kind reminder**
> >
> > Dear reviewer,
> >
> > As the PDF upload deadline is approaching and we have uploaded the latest revised manuscript, we kindly remind you that we are awaiting your response. We sincerely appreciate your efforts and thank you again for your time and consideration.

---

> > > ### Author Response · Authors · 2024-12-03
> > > **Last reminder**
> > >
> > > Dear reviewer,
> > >
> > > We believe that our rebuttal has addressed your concerns.
> > >
> > > As the discussion deadline is approaching, please put your response as soon as possible.
> > >
> > > Our discussion with Reviewer 8SUr during the rebuttal period might help your decision.
> > >
> > > We appreciate your efforts.

---

### Official Review · Reviewer_1XWX · 2024-11-09

**Soundness:** 3
**Presentation:** 3
**Contribution:** 3
**Rating:** 6
**Confidence:** 3

**Summary:**

ConcreTizer, a novel model inversion attack specifically designed for restoring 3D point cloud scenes from disrupted 3D feature data, is presented in this paper.  The unique challenges in 3D point cloud inversion, such as the inherent sparsity of the data and the ambiguity between empty and non-empty voxels after voxelization have been identified.  The authors address these challenges by incorporating two key techniques: Voxel Occupancy Classification (VOC) to distinguish between empty and non-empty voxels, and Dispersion-Controlled Supervision (DCS) to mitigate the dispersion of non-empty voxels across feature extractor layers. Evaluations on widely used 3D feature extractors and benchmark datasets demonstrate the effectiveness of ConcreTizer in restoring original 3D point cloud scenes.  This work highlighted the vulnerability of 3D data to inversion attacks.

**Strengths:**

The paper presents the first in-depth study of model inversion attacks for 3D point cloud data.  The proposed ConcreTizer appears to be novel, designed to tackle the unique challenges associated with 3D point cloud inversion.

The paper is well-written with a clear and concise explanation of the problem, the proposed solution, and the experimental results. The experiments are comprehensive and well-designed, demonstrating the effectiveness of ConcreTizer across various datasets and 3D feature extractors.

The findings of this paper have significant implications for the privacy and security of 3D point cloud data. The authors demonstrate that 3D data is vulnerable to inversion attacks, even when shared in the form of intermediate features.  This highlights the urgent need for robust defense strategies to protect sensitive information in 3D data. The proposed ConcreTizer method serves as a valuable tool for assessing the effectiveness of such defense strategies.

**Weaknesses:**

While the paper emphasizes the need for defense strategies, it only briefly explores the use of Gaussian noise addition as a potential defense mechanism.  A more in-depth investigation of various defense strategies, including their effectiveness and limitations, would further strengthen the paper's contribution.  For example, I would recommend exploring adversarial training, differential privacy, or feature obfuscation techniques. I would recommend an analysis of the trade-offs between privacy protection and utility preservation in these defense mechanisms.

ConcreTizer method is specifically designed for voxel-based feature extractors.  While these extractors are commonly used in autonomous vehicles, exploring the applicability of the method to other types of 3D feature extractors, e.g., point-based and graph-based would help broaden its scope and impact.   I recommend discussing how the key components of ConcreTizer (VOC and DCS) might need to be adapted for point-based or graph-based extractors, and highlight challenges they anticipate in extending the method to these architectures.

While the paper discusses the potential privacy risks associated with 3D point cloud data, it could benefit from a more detailed discussion of the real-world implications of these risks. This could include specific examples of how restored 3D scenes could be used to compromise privacy or security.  I also recommend the authors to provide concrete examples of privacy breaches that could occur with restored 3D scenes, such as identifying individuals, revealing sensitive infrastructure details, or inferring behavioral patterns. It's useful to discuss potential legal or ethical implications of these privacy risks in the context of autonomous vehicles and smart cities.

**Questions:**

Can the authors comment on how to improve the exploration of the defense strategies beyond what is done here? What are other directions and what are inherent challenges?

Any comments on whether the identified model inversion risks apply to point-based and graph-based feature extractors?

Any comments on how the identified privacy risks would compromise privacy or security in more concrete scenarios?  Why would anyone care about this?

---

> ### Author Response · Authors · 2024-11-21
> **Addressing W1 (Q1)**
>
> **W1 (Q1). How to improve the exploration of defense strategies. What are other directions and what are inherent challenges?**
>
> Thank you for your suggestion.
> As you mentioned, there are potential defense mechanisms against inversion attacks.
> 1. **Differential Privacy (DP)**:
> DP provides protection against worst-case privacy leakage through the concept of a privacy budget [1,2]. The budget determines the magnitude of noise to be added, typically in the form of Gaussian or Laplacian noise. However, as shown in our experiments, **adding noise sacrifices utility**, making it unsuitable for accuracy-critical applications like autonomous driving. Recent efforts integrate DP with generative models for anonymization [3,4], but such methods face **latency issues**, limiting real-time usage.
> 2. **Adversarial Training**:
> Adversarial training improves robustness by jointly training with an attack model. Earlier methods [5,6] used generative models for obfuscation, but the associated overhead hinders their practicality in latency-sensitive scenarios. Recent method [7] proposes adversarial training without generative models, relying solely on the utility model. While this reduces complexity, it requires retraining the feature extractor and maintaining both a stand-alone model and a feature-sharing model, increasing system overhead.
> 3. **Feature Obfuscation**:
> Reducing mutual information between raw data and feature data through loss functions [8] can achieve a good privacy-utility balance. However, managing both stand-alone and feature-sharing models adds operational complexity.
>
> Future Directions:
> To defend against inversion attacks in autonomous driving, innovative obfuscation techniques are needed to achieve optimal privacy-utility trade-offs while considering tight latency requirements. We have added a section (F.5) in the supplementary materials that discusses the potential defense mechanisms.
>
>
> [1] M. Abadi et al., "Deep learning with differential privacy," ACM SIGSAC, 2016.
>
> [2] Y. Zhao and J. Chen, "A survey on differential privacy for unstructured data content," ACM Computing Surveys, vol. 54, no. 10s, pp. 1-28, 2022.
>
> [3] J. W. Chen et al., "Perceptual indistinguishability-net (pi-net): Facial image obfuscation with manipulable semantics," IEEE/CVF CVPR, 2021.
>
> [4] H. Xue et al., "DP-image: Differential privacy for image data in feature space," arXiv preprint, 2021.
>
> [5] N. Raval, A. Machanavajjhala, and J. Pan. "Olympus: Sensor privacy through utility aware obfuscation," Privacy Enhancing Technologies, 2019.
>
> [6] Z. Wu et al., "Towards privacy-preserving visual recognition via adversarial training: A pilot study," ECCV, 2018.
>
> [7] S. Liu et al., "Privacy adversarial network: representation learning for mobile data privacy," ACM IMWUT, 2019.
>
> [8] J. Zhang et al., "Privacy-utility trades in crowdsourced signal map obfuscation," Computer Networks, vol. 215, p. 109187, 2022.
>
> ---
> In addition, to confirm the challenges for designing an effective defense method for our attack, we tested popular techniques, rotation, scaling, and random sampling methods on our inversion attack scenario, with the setups below:
>
> * Rotation: Randomly selected from the range of -a to +a degrees (a = 1,2,3,4,5).
>
> * Scaling: Randomly selected from the range of -s to +s percent (s = 2.5,5,7.5,10).
>
> * Random Sampling: Randomly select r% of points (r = 80,60,40, 20,15,10,5).
>
> | Rotation (°) | 0       | 1       | 2       | 3       | 4       | 5       |
> |--------------|---------|---------|---------|---------|---------|---------|
> | **AP**       | 81.7702 | 38.3764 | 17.0799 | 12.0967 | 6.1025  | 2.7797  |
> | **CD**       | 0.0776  | 0.1142  | 0.1728  | 0.2310  | 0.2848  | 0.3344  |
>
>
> | Scaling (%)  | 0       | 2.5     | 5       | 7.5     | 10      |
> |--------------|---------|---------|---------|---------|---------|
> | **AP**       | 81.7702 | 44.2184 | 17.4126 | 9.6410  | 5.2598  |
> | **CD**       | 0.0776  | 0.1690  | 0.2542  | 0.3086  | 0.3516  |
>
>
> | Sampling (%) | 100     | 80      | 60      | 40      | 20      | 15      | 10      | 5       |
> |--------------|---------|---------|---------|---------|---------|---------|---------|---------|
> | **AP**       | 81.7702 | 77.8460 | 77.2418 | 72.5357 | 58.3154 | 52.7130 | 40.3081 | 24.5862 |
> | **CD**       | 0.0776  | 0.0794  | 0.0891  | 0.1080  | 0.1516  | 0.1717  | 0.2034  | 0.2789  |
>
> Similar to the Gaussian Noise method used in the original manuscript, we confirmed that all the three defense methods have the trade-off between utility (Average precision for 3D object detection accuracy) and privacy protection (chamfer distance for point cloud similarity). Experimental results showed that as the intensity of augmentation increased, the drop in AP became significantly larger across all methods. **No method is free from the trade-off between privacy and utility**, which highlights the challenges of defending against inversion attacks in applications like autonomous driving, which demand high accuracy.

---

> ### Author Response · Authors · 2024-11-21
> **Addressing W2 (Q2) and W3 (Q3)**
>
> **W2 (Q2). Whether the identified model inversion risks apply to point-based and graph-based feature extractors**
>
> Thank you for your valuable comment.
>
> As you pointed out, we designed VOC and DCS with a focus on the characteristics of voxel-based feature extractors, such as voxel occupancy and VoI dispersion. However, point-based and graph-based models have different characteristics, requiring tailored design approaches.
>
> In voxel-based models, zero-padding-based representations for empty regions introduce ambiguities, and downsampling via convolution disperses Voxels of Interest (VoI) into neighboring regions. Our VOC and DCS methods mitigate these issues effectively. In contrast, point-based and graph-based models exclusively handle valid points and employ different techniques like Farthest Point Sampling (FPS) for downsampling, resulting in increasingly sparse point clouds. Consequently, **challenges arising from the interaction of empty and non-empty regions in voxel-based models are absent in these architectures. Instead, addressing sparsity through effective point upsampling techniques becomes critical [1].**
>
> While the **progressive restoration concept in DCS has the potential** for broader applicability across different data representations, we believe that point- and graph-based models require bespoke attack methods tailored to their unique characteristics.
>
>
> [1] Y. He et al., "Grad-pu: Arbitrary-scale point cloud upsampling via gradient descent with learned distance functions," IEEE/CVF CVPR, 2023.
>
>
>
> ---
> **W3 (Q3). How the identified privacy risks would compromise privacy or security in more concrete scenarios.**
>
> Privacy risks associated with restored 2D images are well-recognized, but similar concerns in 3D point clouds are often underestimated. In our work, we identified specific private information that can be leaked from 3D point clouds (lines 39-46) and evaluated 3D object detection performance using the PointPillars model.
>
> We agree that including a wider range of downstream tasks would enhance the evaluation. However, **the lack of labels for these tasks in the open-source autonomous driving dataset limited our ability to explore them**. Despite this, we hope our study serves as a starting point for further research into the privacy implications of 3D point cloud data in autonomous vehicle applications. Future work could extend this foundation to evaluate the impact of such attacks on additional downstream tasks and scenarios.

---

> > ### Author Response · Authors · 2024-12-03
> > **Reminder**
> >
> > Dear reviewer,
> >
> > We believe that our rebuttal has addressed your concerns.
> >
> > As the discussion deadline is approaching, please put your response as soon as possible.
> >
> > Our discussion with Reviewer 8SUr during the rebuttal period might help your decision.
> >
> > We appreciate your efforts.

---

### Author Response · Authors · 2024-11-21
**Global response to all the reviewers**

Dear Reviewers,

We sincerely thank you for taking the time to review our work and providing constructive feedback.

We have reviewed the comments below and will ensure that all feedback is incorporated into the final version of the paper.
* Reviewer (1XWX), Reviewer (3vLL), and Reviewer (VptB) acknowledged the originality of our study as the first to address inversion attacks on 3D point cloud data.
* Reviewers (1XWX) and (VptB) particularly emphasized significant implications of our research from a privacy perspective.
* Reviewer (1XWX), Reviewer (VptB), and Reviewer (8SUr) appreciated the well-organized writing of the paper, with Reviewer (8SUr) specifically commending the figures for effectively visualizing the problem's root cause and clearly illustrating the proposed method.
* Reviewers (1XWX) and (3vLL) acknowledged the comprehensive and well-executed evaluation, noting that it convincingly demonstrates the superiority of our proposed method.

We will provide detailed responses to the weaknesses (W) and questions (Q) in the respective rebuttal tabs for each reviewer.

The revised parts based on your feedback are marked in red in both the main text and the supplementary materials. Your attention to these revisions would be greatly appreciated.

---

### Meta-Review · Area_Chair_RB3f · 2024-12-22

**Metareview:**

This work proposed a novel model inversion attack specifically designed for restoring 3D point cloud scenes from disrupted 3D feature data. The submission identifies the challenges of model inversion attack for voxel-based 3D point cloud feature extractions and addresses these challenges by formulating it as the voxel occupancy classification task to distinguish between empty and non-empty voxels with the Dispersion-Controlled Supervision (DCS) to mitigate the dispersion of non-empty voxels across feature extractor layers. Evaluations on two widely used 3D feature extractors and benchmark datasets demonstrate the effectiveness of ConcreTizer in restoring original 3D point cloud scenes. From the first round of reviewing, reviewers agree that: (1) The submission presents the first in-depth study of model inversion attacks for 3D point cloud data.  (2) The paper is well-written. (3) The study is important to highlight the privacy and security issues of 3D point cloud data. Reviewers also have a lot of concerns: (1) The narrow study on voxel-based feature extraction methods. (2) Unclear motivations for proposing a new method. (3) Limited validation scope. Two reviewers gave positive scores while another two provided borderline negative rates. During the rebuttal process, only one reviewer joined the discussion and provided meaningful comments to the authors.  I reviewed the concerns of the reviewers, the authors' responses, and the revised submission. I agree this work proposed a novel attack for the voxel feature-based 3D point cloud models, and the results validate the effectiveness of the methods. Based on the revised version, this work could be accepted. However, the concerns and additional results in the rebuttal should be included in the final version. In addition, this work paid more attention to the 3D point cloud restoration instead of the model inversion attack task in the introduction and method sections, which confuses readers of the main objective. Luckily, the revision involves the results of using the generated data to train 3D object detectors. This part should be further enhanced in the final version.

**Additional Comments On Reviewer Discussion:**

From the first round of reviewing, reviewers agree that: (1) The submission presents the first in-depth study of model inversion attacks for 3D point cloud data.  (2) The paper is well-written. (3) The study is important to highlight the privacy and security issue of 3D point cloud data. Reviewers also have a lot of concerns: (1) The narrow study on voxel-based feature extraction methods. (2) Unclear motivations for proposing a new method. (3) Limited validation scope. Two reviewers gave positive scores while another two provided borderline negative rates. During the rebuttal process, only one reviewer joined the discussion and provided meaningful comments to the authors.  I reviewed the concerns of the reviewers, the authors' response, and the revised submission. I agree this work proposed a novel attack for the voxel feature-based 3D point cloud models, and the results validate the effectiveness of the methods. Based on the revised version, this work could be accepted. However, the concerns and additional results in the rebuttal should be included in the final version. In addition, this work paid more attention to the 3D point cloud restoration instead of the model inversion attack task in the introduction and method sections, which confuses readers of the main objective. Luckily, the revision involves the results of using the generated data to train a 3D object detectors. This part should be futher enhanced in the final version.

---

> ### Public Comment · ~Youngseok_Kim2 · 2025-02-26
> **Response to AC Comment**
>
> Thank you for your valuable feedback.
>
> * We have incorporated all the results from the rebuttal process, including 3D object detection performance across various models, into the camera-ready version.
>
> * Inversion attack refers to the process of recovering the original data from the extracted features, meaning that point cloud restoration and inversion attack is tightly coupled. To clarify this, we have added a definition of the inversion attack in the introduction (Section 1) and provided more detailed explanations in the problem definition (Section 4.2).
>
> * Regarding the evaluation of 3D object detection models (Section 5.3), we would like to clarify that the restored data was used exclusively for inference and not for training. To eliminate any ambiguity, we have updated the phrase “separately trained model” to “pre-trained model.”
>
> Once again, thank you for your insightful comments.

---

### Decision · Program_Chairs · 2025-01-22

Accept (Poster)